



Geoscientific
Model Development

# A Gaussian process emulator for simulating ice sheet–climate interactions on a multi-million-year timescale: CLISEMv1.0

**Jonas Van Breedam**[1], **Philippe Huybrechts**[1], **and Michel Crucifix**[2]

[1]Earth System Science & Departement Geografie, Vrije Universiteit Brussel, Brussels, Belgium
[2]Earth and Life Institute, UCLouvain, Louvain-la-Neuve, Belgium

**Correspondence:** Jonas Van Breedam (jonas.van.breedam@vub.be)

**Abstract.** On multi-million-year timescales, fully coupled ice sheet–climate simulations are hampered by computational limitations, even at coarser resolutions and when using asynchronous coupling schemes. In this study, a novel coupling method CLISEMv1.0 (CLimate–Ice Sheet EMulator version 1.0) is presented, where a Gaussian process emulator is applied to the climate model HadSM3 and coupled to the ice sheet model AISMPALEO. The temperature and precipitation fields from HadSM3 are emulated to feed the mass balance model in AISMPALEO. The sensitivity of the evolution of the ice sheet over time is tested with respect to the number of predefined ice sheet geometries that the emulator is calibrated on. Additionally, the model performance is evaluated in terms of the formulation of the ice sheet parameter (being ice sheet volume, ice sheet area or both) and the coupling time. Sensitivity experiments are conducted to explore the uncertainty introduced by the emulator. In addition, different lapse rate adjustments are used between the relatively coarse climate model and the much finer ice sheet model topography. It is shown that the ice sheet evolution over a million-year timescale is strongly sensitive to the definition of the ice sheet parameter and to the number of predefined ice sheet geometries. With the new coupling procedure, we provide a computationally efficient framework for simulating ice sheet–climate interactions on a multi-million-year timescale that allows for a large number of sensitivity tests.

## 1 Introduction

Earth system models provide the state-of-the-art method for quantifying feedbacks between the different components of the climate system on a decadal to centennial timescale (Eyring et al., 2016). On millennial to multi-millennial timescales, Earth system models of intermediate complexity are used to explore the feedbacks in the climate system between the ice sheets, the atmosphere and the ocean (Eby et al., 2013; Van Breedam et al., 2020). Those fully coupled models, even at coarser resolution, are computationally very expensive, and other techniques have been proposed to simulate ice sheet–climate interactions on a (multi-)million-year timescale.

The basic asynchronous method, also called the direct asynchronous method, is a simple and straightforward coupling technique to address the different response times between ice sheets and the atmosphere (Pollard, 2010). The direct asynchronous method consists of running a climate model, typically a general circulation model (GCM), for a few decades until steady state and then using the relevant climatic information over the polar regions as input to an ice sheet model. The ice sheet model is typically run for a few thousand years before the climatic information is updated (e.g. DeConto and Pollard, 2003; Gasson et al., 2016). This procedure is repeated for the entire time span of interest. The indirect asynchronous coupling, also called the matrix method or the GCM lookup table (Pollard, 2010), is based on predefined, idealized GCM snapshots that span the possible forcing during the entire simulation period. The matrix lookup table is more sophisticated, with the creation of a matrix of climate model output from the extremes of the forcing

and some intermediate climate states in between (Ladant et al., 2014; Stap et al., 2017; Berends et al., 2018). The coupling procedure linearly interpolates the climatic fields from precursor climate model runs.

An alternative approach is to consider a Gaussian process emulator. A Gaussian process emulator is a statistical model that fits a Gaussian process to data in order to link input with output fields of a model, generally referred to as the simulator (Andrianakis and Challenor, 2012). Emulators have been used for a number of applications in climate science, for instance as a tool to predict the future climate evolution (Levermann et al., 2020) or sea level rise as a result of land ice melting (Edwards et al., 2021), based on large ensembles of simulations, each with different model input parameters. It is also a useful technique to couple different components of the climate system that would require large computational resources, such as an atmosphere–ocean coupling (Tran et al., 2019). An emulator has been used to assess the sensitivity of the climate during the Pleistocene (Araya-Melo et al., 2015) and the late Pliocene (Lord et al., 2017). In these simulations, the ice sheets are static and defined by different ice sheet geometries. So far, an emulator has not been used to study the climate system including dynamic ice sheets.

Here a Gaussian process emulator is presented that is calibrated on the climatic output from the climate model HadSM3 to force an ice sheet model in order to predict the climate over Antarctica during the late Eocene. The late Eocene to Eocene–Oligocene transition is chosen because of the large contrast in continental glaciation and large variations in climate forcing, such as $CO_2$ concentrations, at this time (Pagani et al., 2011; Zhang et al., 2013). The ice sheet model runs continuously over a multi-million-year period and passes the ice sheet geometry information (referred to as the ice sheet parameter) to the emulator (statistical representation of the climate model). The emulator calculates the climatic variables based on the prescribed external forcing (carbon dioxide concentration and orbital parameters) and the actual ice sheet parameter and returns temperature and precipitation data to the ice sheet model. This coupling procedure is novel, but various implementation choices may influence the result: the approach for lapse rate adjustment, the coupling time between ice sheet and climate, and the definition of emulator input variables. In addition, the number of GCM experiments on which the emulator is tuned might have an influence on the predicted climate. The key questions to be answered can be summarized as follows.

1. The ice sheet parameter is defined by a number that represents the influence of the ice sheet in the climate system (Araya-Melo et al., 2015; Lord et al., 2017). The ice sheet mainly influences the local climate via its distinct albedo, its height and its freshwater input into the ocean (not taken into account in this study). Therefore, it is not trivial to determine how the ice sheet parameter should be defined as a single number. Is ice volume a proper way to define the ice sheet parameter? Does ice area represent the climatic changes better? Is it best to calibrate the emulator based on both ice volume and ice area?

2. The emulator needs a number of input ice sheet geometries to simulate the climate for a range in orbital parameters and $CO_2$ values for the given ice sheet geometry. How many ice sheet geometries and climate model experiments are needed? Does the spacing between different ice sheet geometries influence the model performance?

3. The lapse rate adjustment between a coarse-resolution climate model and the high-resolution ice sheet model is usually applied by a constant value for the moist adiabatic lapse rate over the domain. Common values are $5\,°C\,km^{-1}$ (Ladant et al., 2014), $6.5\,°C\,km^{-1}$ (Löfverström et al., 2015), $7\,°C\,km^{-1}$ (Thompson and Pollard, 1997) or $8\,°C\,km^{-1}$ (Berends et al., 2018). The lapse rate above ice-covered regions is found to be $4.9\,°C\,km^{-1}$, smaller than the typical values for the moist adiabatic lapse rate (Gardner et al., 2009). Moreover, the near-surface lapse rate varies spatially and temporarily between diurnal and seasonal cycles as opposed to the free adiabatic lapse rate that has a rather constant value (Marshall et al., 2006). What is the influence of using a different lapse rate on the ice sheet evolution?

4. With asynchronously coupled climate–ice sheet model runs, given the long response time of the ice sheets and the computational limits, one generally only updates the climatic information every several thousand years. However, the choice of the coupling time might have an influence on the ice sheet evolution over time. What is the optimal coupling time to have a realistic, yet efficient, model running?

5. What is the uncertainty introduced by the emulator, and what is its influence on the coupled ice sheet–climate simulations?

## 2  Model description

In this section, the new coupling method CLISEMv1.0 is described together with the climate model HadSM3 and the ice sheet model AISMPALEO. CLISEMv1.0 is calibrated on climatic output from HadSM3 and provides the forcing fields (monthly temperature and precipitation) for the ice sheet model. The ice sheet model AISMPALEO returns the ice sheet volume and/or area to CLISEMv1.0.

## 2.1 Climate model HadSM3

The climate model HadSM3 (Williams et al., 2001) is an atmosphere–slab ocean general circulation model (GCM). It has a resolution of 2.5° in latitude and 3.75° in longitude, with 19 levels in the vertical (Gordon et al., 2000). The MOSES-1 scheme is chosen as the land surface scheme (Cox et al., 1999) with a tundra-like albedo on the Antarctic continent where no ice is present and an albedo for snow where ice is present. Sea surface temperatures are reconstructed based on a best estimate from Evans et al. (2017) for the late Eocene in order to calibrate the corrective heat fluxes from the slab ocean model. These corrective heat fluxes represent the seasonal deep-water exchange and horizontal heat transport that is present in the real ocean. The oceanic heat fluxes are exchanged between the atmosphere and the slab ocean model in the mixed layer, which is 50 m thick in our simulations. In this way, realistic sea surface temperatures are simulated for the different climate model simulations.

The model is chosen because of its good performance over the Antarctic ice sheet for the present day and the Last Glacial Maximum (Connolley and Bracegirdle, 2007; Maris et al., 2012). Moreover, HadSM3 is a computationally efficient climate model that allows for performing a large number of experiments (Valdes et al., 2017). The paleogeographic reconstruction for the simulations is based on the method presented in Baatsen et al. (2016) and makes use of the GPlates software (Müller et al., 2018). The paleogeographic reconstruction represents the continental configuration at 39 Ma as representative for the late Eocene. As a result, the Antarctic continent has a slightly different position compared to the present-day. The bedrock topography for Antarctica is derived from the Wilson et al. (2012) maximum bedrock elevation reconstruction (Fig. 1). Simulated temperatures for a warm orbital configuration (maximum austral summer insolation) and a cold orbital configuration (minimum austral summer insolation) are shown in Fig. 2a and b, respectively.

## 2.2 Antarctic ice sheet model AISMPALEO

The Antarctic ice sheet model AISMPALEO is a three-dimensional thermomechanical ice sheet and ice shelf model (Huybrechts and de Wolde, 1999) used for simulating ice sheet dynamics during periods in the past (Goelzer et al., 2016a, b) and the future (Seroussi et al., 2020; Van Breedam et al., 2020). The Shallow Ice Approximation is used to calculate the grounded ice flow, which is a result of internal deformation and basal sliding where the pressure melting point is reached. The model comprises a component taking into account the solid Earth response due to ice loading. This component consists of a rigid elastic plate on top of a viscous asthenosphere to allow for deviations from local isostatic loading. The surface mass balance is computed using the positive degree day (PDD) method (Janssens and Huybrechts,

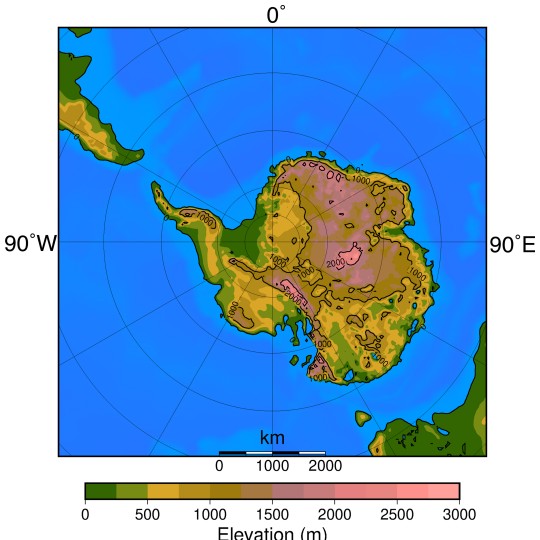

**Figure 1.** Antarctic bedrock topography following Wilson et al. (2012) as used in the simulations. Latitudes are given every 10° and longitudes every 30°. Note the different paleogeographic position of the continents from today.

2000), where the yearly sum of daily average temperatures above 0 °C is used to determine the melt potential. The standard deviation of the mean daily temperature is 4.2 °C, representing random weather fluctuations and the daily cycle. The difference in snow and ice albedo in the ice sheet model is taken into account by using a PDD factor for snow melting of 0.003 m ice equivalent (i.e.) °C$^{-1}$ d$^{-1}$ and a PDD factor for ice melting of 0.008 m i.e. °C$^{-1}$ d$^{-1}$. Monthly mean temperature and precipitation are used from HadSM3 to drive the PDD model. The rain limit is chosen at 1 °C and determines whether precipitation falls as snow or as rain. Meltwater retention allows runoff to be retarded and/or to eventually refreeze in the snowpack. Ice shelf formation is included and calculated using the shallow shelf approximation. Ice shelves start to form when the grounding line reaches the coast and the influx of ice from the continent exceeds the ablation (surface ablation and basal melting). A constant basal melt rate of 1 m yr$^{-1}$ is used in all the simulations. The ice sheet model is run at a resolution of 40 km to allow for long integrations.

## 2.3 CLISEMv1.0: set-up and calibration

The Gaussian process (GP) principal component analysis (PCA) emulator (Wilkinson, 2010; Bounceur et al., 2015; Lord et al., 2017) used in this study is a statistical representation of the climate model HadSM3 (the simulator). The emulator is calibrated on a relatively small number of climate model runs and aims to predict the climate for any combination of climatic forcing of the original climate model runs. To allow for reliable predictions, the initial climate model runs need to fill the entire multi-dimensional input space. In our

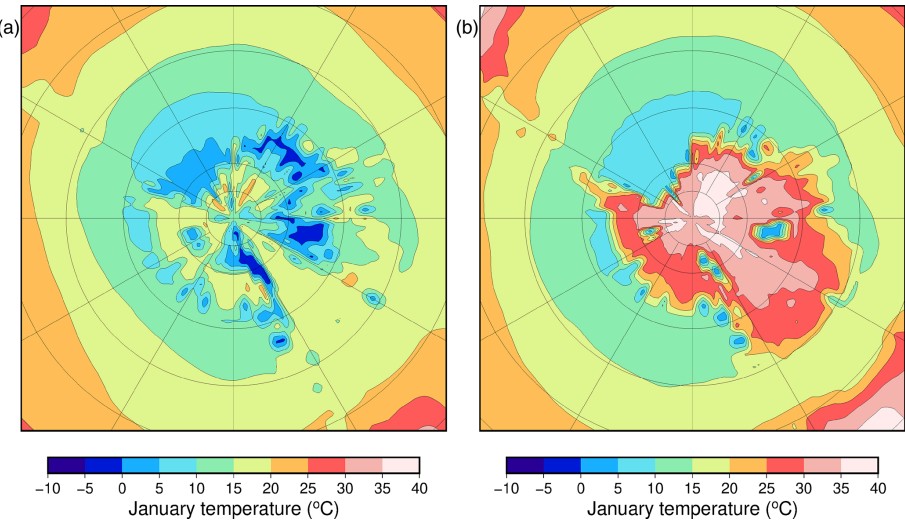

**Figure 2.** Simulated January mean surface air temperature (°C) for **(a)** a cold orbital configuration and **(b)** a warm orbital configuration for a $3\times CO_2$ scenario (840 ppmv).

case, the multi-dimensional input space consists of the orbital parameters (eccentricity $e$, obliquity $\varepsilon$ and longitude of perihelion $\varpi$), the carbon dioxide concentration forcing, and the ice sheet size and extent defined by the ice sheet parameter. The theoretical background has already been discussed in detail in previous papers (Araya-Melo et al., 2015; Bounceur et al., 2015; Lord et al., 2017), and here the focus is on the implementation of the dynamic ice sheet component.

It is recommended to have at least 10 experiments per input parameter (Loeppky et al., 2009). Therefore, the recommended minimum number of experiments with our five input parameters (three orbital parameters, $CO_2$ concentration and ice sheet parameter) would be 50. Since the atmosphere–slab ocean model is time efficient, we have chosen to run 100 climate model runs with five variable forcing parameters. The ice sheets have a very distinct climatic imprint compared to the orbital parameters and the $CO_2$ level, which all result in smooth climatic fields. Because of the large difference in albedo between ice and tundra at the edge of the ice sheet, the climatic imprint of a certain ice sheet geometry has a sharp boundary. The number of ice sheet geometries taken into the model design of the emulator might therefore have a large impact on the performance of the emulator. To test the impact of the ice sheet parameter, four different emulators are constructed based on a different number of predefined ice sheet geometries or based on a different spread of the ice sheet geometries (Figs. 3, A2, A3 and A4). The different emulators are named according to the number of ice sheet geometries in the model design: 8, 12, and 20 for EMULATOR_8, EMULATOR_12a and EMULATOR_12b, and EMULATOR_20, respectively.

Except for the number of predefined ice sheet geometries, the spread of the different prescribed geometries also varies between the different emulators, depending on whether the

ice sheet parameter is defined by ice area or by ice volume. EMULATOR_8, EMULATOR_12a and EMULATOR_20 have a good spread between the different ice sheet geometries in terms of ice volume and ice area. EMULATOR_12b is well defined for ice volume but poorly defined by ice sheet area as there are several experiments with the same ice sheet area but different ice sheet geometry (Fig. 4 and Table A1 in Appendix A for the experimental parameter values). In this way, the influence of the spread in ice sheet volume and ice area on the emulated climate is investigated. The spacing of the different ice sheet geometries is expected to be crucial for medium-sized ice sheets because they constitute a transition zone towards a fully glaciated continent. EMULATOR_20 has the smallest spacing of ice sheet geometries around the crucial medium-sized ice sheets, separated at the minimum distance that corresponds to the resolution of the climate model. The maximum ice sheet geometry in the model design for EMULATOR_12 is smaller than the maximum ice sheet geometry in the model design for EMULATOR_20 and EMULATOR_8. The objective of designing EMULATOR_12 is to evaluate to what extent the emulator can still be used in an extrapolation regime beyond the largest ice sheet geometry.

A model design with 100 GCM experiments is constructed where each experiment has a different combination of orbital parameters, $CO_2$ concentration values and ice sheet geometry (see Table A1 in Appendix A). Insolation values are well approximated as a linear combination of the eccentricity and longitude of perihelion (Loutre, 1993), and therefore the terms $e\sin\varpi$ and $e\cos\varpi$ in combination with the obliquity $\varepsilon$ are used for the orbital parameter variation in the model design. The range of orbital parameters is taken from Laskar et al. (2004) for the period 40 to 33 Ma. The eccentricity has a maximum value during the period between 40 and 33 Ma of 0.063, and the obliquity is sampled in the range of

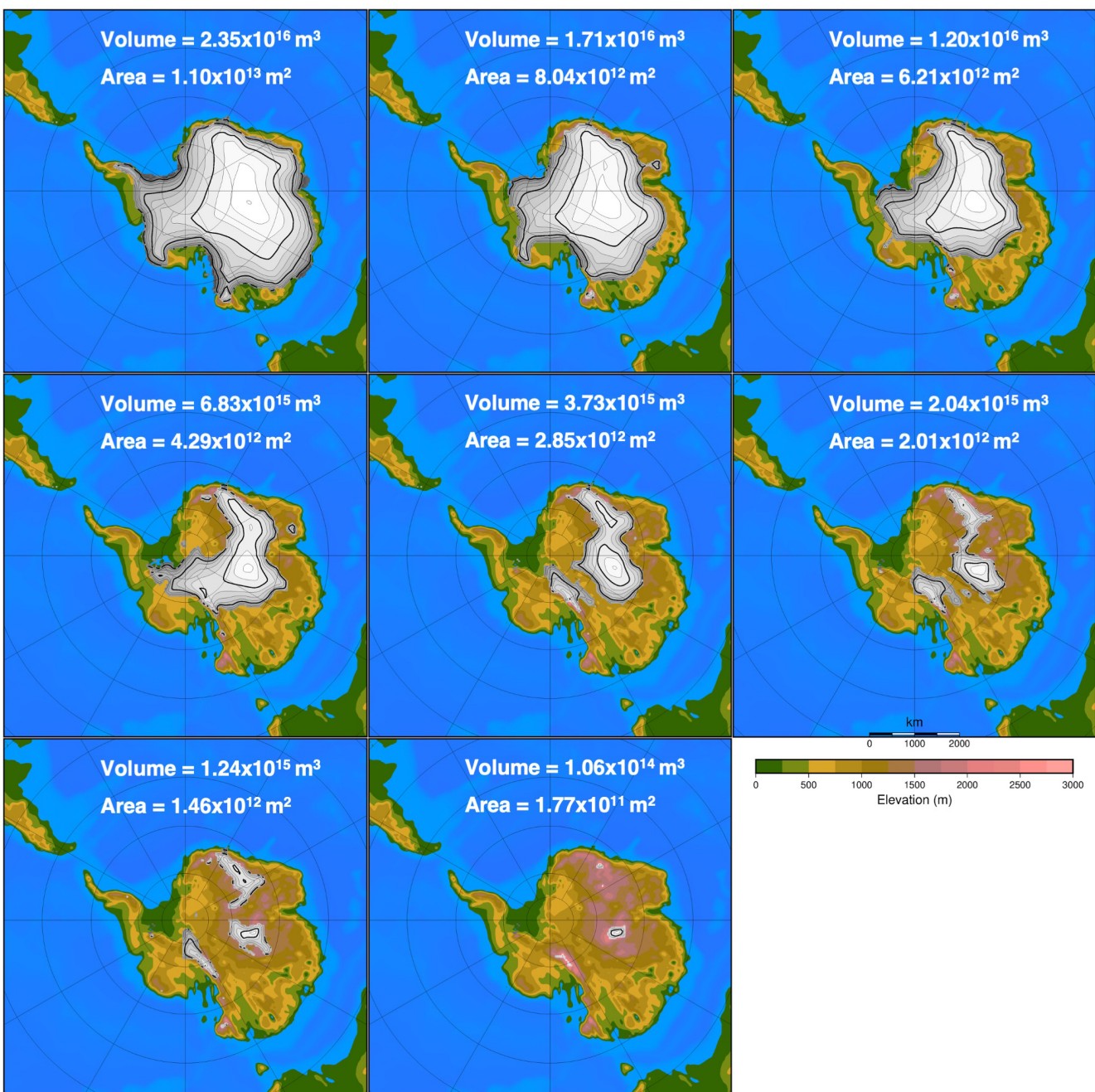

**Figure 3.** Eight different ice sheet geometries with their respective ice sheet volume and ice sheet area as input to EMULATOR_8. Ice sheet contour lines are given every 250 m and thick contour lines every 1000 m.

22–24.5°. The $CO_2$ interval ranges from 550 to 1150 ppmv, roughly equivalent to $2\times CO_2$ to $4\times CO_2$. The ice sheet parameter consists of 8, 12 or 20 predefined ice sheet geometries. They are constructed based on preliminary steady-state ice sheet model runs for a range of different climatic forcings (for EMULATOR_12a and EMULATOR_12b) or from ice sheet geometry snap shots during the build-up of a continental scale ice sheet (for EMULATOR_8 and EMULA-TOR_20). The final ice sheet geometries are chosen to provide a range from an almost ice-free Antarctic continent up to a fully glaciated continent. Tundra is present between the ice sheet margin and the coast. The parameter combinations are constructed using a Latin hypercube design where the minimum Euclidean distance between two parameter combinations is maximized (Fig. 5). With this model design, 100

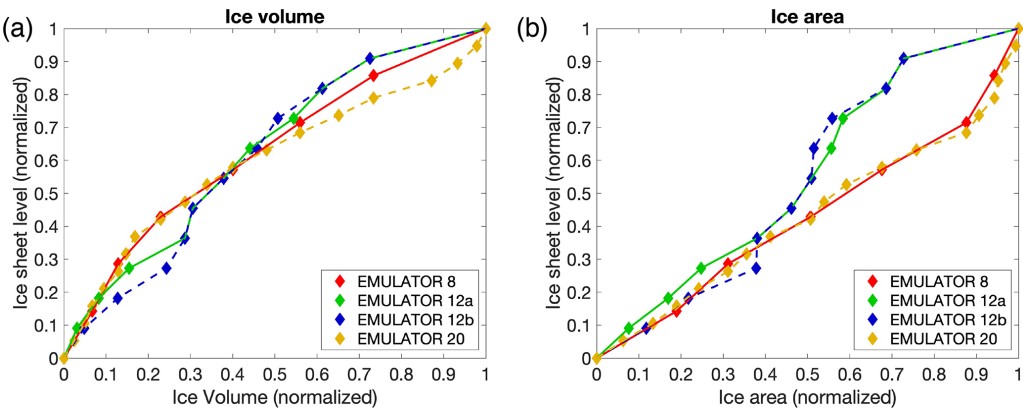

**Figure 4.** Spread of the ice sheet parameter defined by ice volume and ice area for the four different emulators. Note that EMULATOR_8 has only 8 predefined ice sheet geometries, EMULATOR_12a and EMULATOR_12b have 12, and EMULATOR_20 has 20.

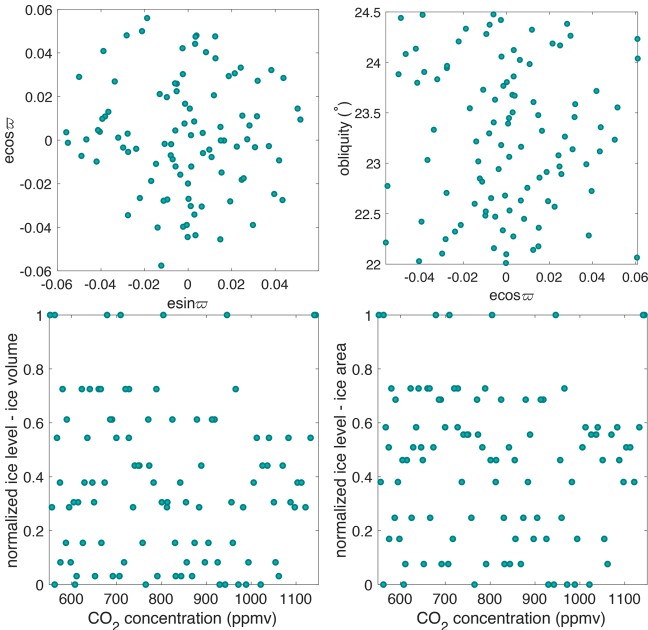

**Figure 5.** Latin hypercube model design of EMULATOR_12a showing the values for the orbital forcing, carbon dioxide forcing and ice sheet parameter forcing (defined by ice volume and ice area). Note that each dot represents one experiment from a total of 100 experiments.

in the matrix $\mathbf{Y}$. Each column of $\mathbf{Y}$ contains the output for one experiment. Here, the matrix $\mathbf{Y}$ only contains climatic output data on the ice sheet model grid ($201 \times 201$ grid points) because our interest is the climate evolution over Antarctica.

The climatic output is modelled as a Gaussian process, defined by a mean function $m(\boldsymbol{x})$ and a covariance function $V(\boldsymbol{x}, \boldsymbol{x}')$. The prior mean function is defined by a linear combination of a set of linear regression functions (Eq. 1), where $\boldsymbol{\beta}$ is a vector of regression coefficients corresponding to the mean function and $h(\boldsymbol{x})$ is a vector of known regression functions of the inputs. The covariance function consists of the correlation function and the scaling value $\sigma^2$ (Eq. 2). The squared exponential correlation function is chosen with the inclusion of the so-called nugget $\nu$ and correlation length $\delta$ hyperparameters (Eq. 3). The nugget was originally meant to deal with sampling variability of the simulator output, but even when this sampling variability is small, it is effective to prevent numerical instability and to compensate for inadequate correlation priors (Andrianakis and Challenor, 2012; Araya-Melo et al., 2015). The correlation length $\delta$ can be understood as a measure of how quickly the model output is changing as a function of the input (Wilkinson, 2010). The larger the distance between two input vectors, the quicker the correlation goes to zero. TS2

$$m(\boldsymbol{x}) = h(\boldsymbol{x})^T \boldsymbol{\beta} \tag{1}$$

$$V\left(\boldsymbol{x}, \boldsymbol{x}'\right) = \sigma^2 \left[c\left(\boldsymbol{x}, \boldsymbol{x}'\right)\right] \tag{2}$$

$$c\left(\boldsymbol{x}, \boldsymbol{x}'\right) = \exp\left\{\sum_{i=1}^{p} \left(\frac{d_i}{\delta_i}\right)^2\right\} + \nu I \tag{3}$$

TS3 The formalism followed here is Bayesian, and the prior mean and prior covariance functions (Eqs. 1–3) are updated (Eqs. 4–5) based on the climate model output data. All values of $\boldsymbol{\beta}$ are a priori equally probable, and we assume a vague conjugate prior $(\boldsymbol{\beta}, \sigma^2)$ that is proportional to $\sigma^2$. The posterior distribution of the model data (temperature and precipitation) is a Student's $t$ distribution with $n - q$ degrees of

GCM runs are performed until the climate (atmosphere and slab ocean) is in steady state with the forcing (40 years).

The design matrix of input data $\mathbf{D}$ ($n \times p$) has 100 rows (number of experiments) and five ($e \sin \varpi$, $e \cos \varpi$, $\varepsilon$, $CO_2$, ice volume or ice area) or six columns ($e \sin \varpi$, $e \cos \varpi$, $\varepsilon$, $CO_2$, ice volume, ice area). Each simulation performed by the climate model is characterized by a row of matrix $\mathbf{D}$, called the input vector $\boldsymbol{x}_i$. The climatic output (temperature and precipitation) where HadSM3 is a function of the input vector is called $f(\boldsymbol{x}_i)$ TS1 is from all 100 experiments saved

freedom (which is close to Gaussian) with a mean $m^*(\boldsymbol{x})$ and covariance $V^*(\boldsymbol{x}, \boldsymbol{x}')$ as follows: TS4

$$m^*(\boldsymbol{x}) = h(\boldsymbol{x})^T \hat{\boldsymbol{\beta}} + t(\boldsymbol{x}) A^{-1} \left( \boldsymbol{y} - H \hat{\boldsymbol{\beta}} \right), \tag{4}$$

$$V^*(\boldsymbol{x}, \boldsymbol{x}') = \frac{\left( \boldsymbol{y} - H \hat{\boldsymbol{\beta}} \right)^T A^{-1} (\boldsymbol{y} - H \hat{\boldsymbol{\beta}})}{n - q - 2}$$
$$\times \left[ \left( c\left( \boldsymbol{x}, \boldsymbol{x}' \right) + \nu I \right) - t(\boldsymbol{x})^T A^{-1} t\left( x' \right) \right.$$
$$\left. + P(\boldsymbol{x}) \left( H^T A^{-1} H \right)^{-1} P(x')^T \right]. \tag{5}$$

A PCA is performed on the climatic output, and between 17 and 20 components are kept before calibrating the emulator (see model code for principal components, length scales and nugget values). The Gaussian process model with the posterior means and variances is applied to each principal component. Once the PCA emulator is calibrated (by optimizing the nugget and length scales), the scores of each principal component can be estimated for arbitrary input values, with an associated covariance that effectively measures the prediction uncertainty.

As described above, four different emulators are constructed (EMULATOR_8, EMULATOR_12a, EMULATOR_12b and EMULATOR_20), and each emulator is calibrated with either ice volume, ice area, or with both ice volume and ice area as the ice sheet parameter. This gives a total of $4 \times 3 = 12$ distinct calibrated emulators to simulate the ice sheet evolution. Calibration of an emulator is achieved by adjusting the length scales $\delta$, the nugget $\nu$ (uncertainty band) and the number of principal components (PCs) in order to minimize the root-mean-square error between the simulated and predicted climatic fields in leave-one-out experiments. It is assumed that the sampling error introduced by model variability is almost negligible because we are using a slab ocean climate model version where the internal climate variability is small and the climate states quickly converge to a mean value. Therefore, the adopted nugget is chosen to be 0.001 (small non-zero uncertainty band around the data) for the temperature and 0.01 for the precipitation emulation. The emulator calibration is done for precipitation and temperature data for each month. During the calibration process, the number of PCs is varied between 5 and 25. It is chosen to keep 20 PCs to explain the observed variation in temperature and 17 PCs to explain the variation in precipitation, since these numbers gave the best emulator performance (as explained below).

R's Nelder–Mead optimization function (Nelder and Mead, 1965) is used to maximize the likelihood of the emulator (Kennedy and O'Hagan, 2000; Lord et al., 2017). We obtain a low length scale for the ice sheet parameter (between 0.02 and 0.5). When looking at the summer temperature for all GCM runs, there is a smooth, almost linear dependency between the ice sheet parameter and the simulated summer temperatures (Fig. 6), and therefore we increased the length

scale for the ice sheet parameter. The optimization function is used to get the correlation lengths for the orbital parameters and the $CO_2$ forcing right and the correlation length for the ice sheet parameter was manually chosen to be 1.2 for all emulators.

The emulator performance is determined by the variance of the emulator and the reliability of the emulator. The variance is a measure of the uncertainty of the mean predictions of the emulator. The reliability of the emulator determines how well the emulator is calibrated (how well the emulator estimates its own uncertainty). Ideally, the emulator is well calibrated and the uncertainty is low. The calibration is investigated by leave-one-out experiments where the simulated temperature is predicted based on the calibrated emulator while leaving out one experiment at a time. The results are visualized in Fig. 7 where the number of grid points that is predicted within 1 (grey) to 4 (red) standard deviations from the simulated temperature is given for each of the GCM runs for EMULATOR_20. Overall, all emulators perform well, since $> 68\,\%$ of the grid points are predicted within 1 standard deviation (Table 1). The calibration based on ice volume and ice area separately shows a very similar reliability and uncertainty. Even though the variance for the emulators calibrated on both ice area and on ice volume is lower, it has a lower reliability because it struggles to capture the output dependency on both variables simultaneously.

The spatial difference between the simulated and the predicted climatic fields is visualized in Fig. 8. January temperatures are shown for experiment *xaemdk*, which performs poorly, and for experiment *xaemdv*, which performs quite well. These experiments are run with the second smallest and smallest ice sheet geometry, respectively. The predicted temperatures are warm-biased up to $8\,°C$ for the tundra region and cold-biased up to $10\,°C$ for the ice-covered region for experiment *xaemdk*. The bias is much smaller for experiment *xaemdv* with errors of less than $2\,°C$ over most of the Antarctic continent. Other experiments that perform poorly, such as *xaemdg* and *xaembb*, have a very high eccentricity, high obliquity, and a summer during aphelion or perihelion. They have the most extreme insolation values and lay at the edge of the experimental design, which may explain why the emulator does a poorer job of predicting the simulated temperatures.

## 2.4 Coupling procedure between AISMPALEO and the emulator

Due to computational limitations, the coupling procedure between ice sheets and climate on a multi-million year timescale is always asynchronous. However, the emulator–ice sheet coupling leaves the possibility for a very short coupling time because once the emulator is calibrated, it can be run in stand-alone modus (without the need to use the simulator). In these simulations, the ice sheet model is initialized from an ice-free state. After a predefined time (1000 years

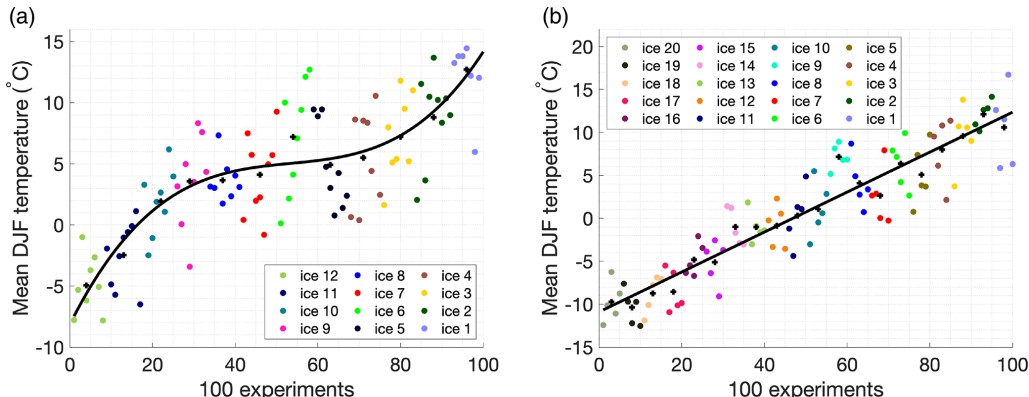

**Figure 6.** Mean austral summer temperature (December, January, February) for **(a)** EMULATOR_12a and **(b)** EMULATOR_20. Each dot represents the output from one GCM run. The output is grouped for each input ice sheet geometry. The mean austral summer for each input ice sheet geometry is given by the black cross, and the best fit is given by the black line.

**Figure 7.** Calibration of EMULATOR_20 for the ice sheet parameter defined by the ice volume **(a)**, the ice area **(b)**, and both ice volume and ice area **(c)**. The bars indicate the percentage of grid points where the emulator predicts the January temperature above the Antarctic continent within 1, 2, 3 or 4 standard deviations for each of the 100 experiments.

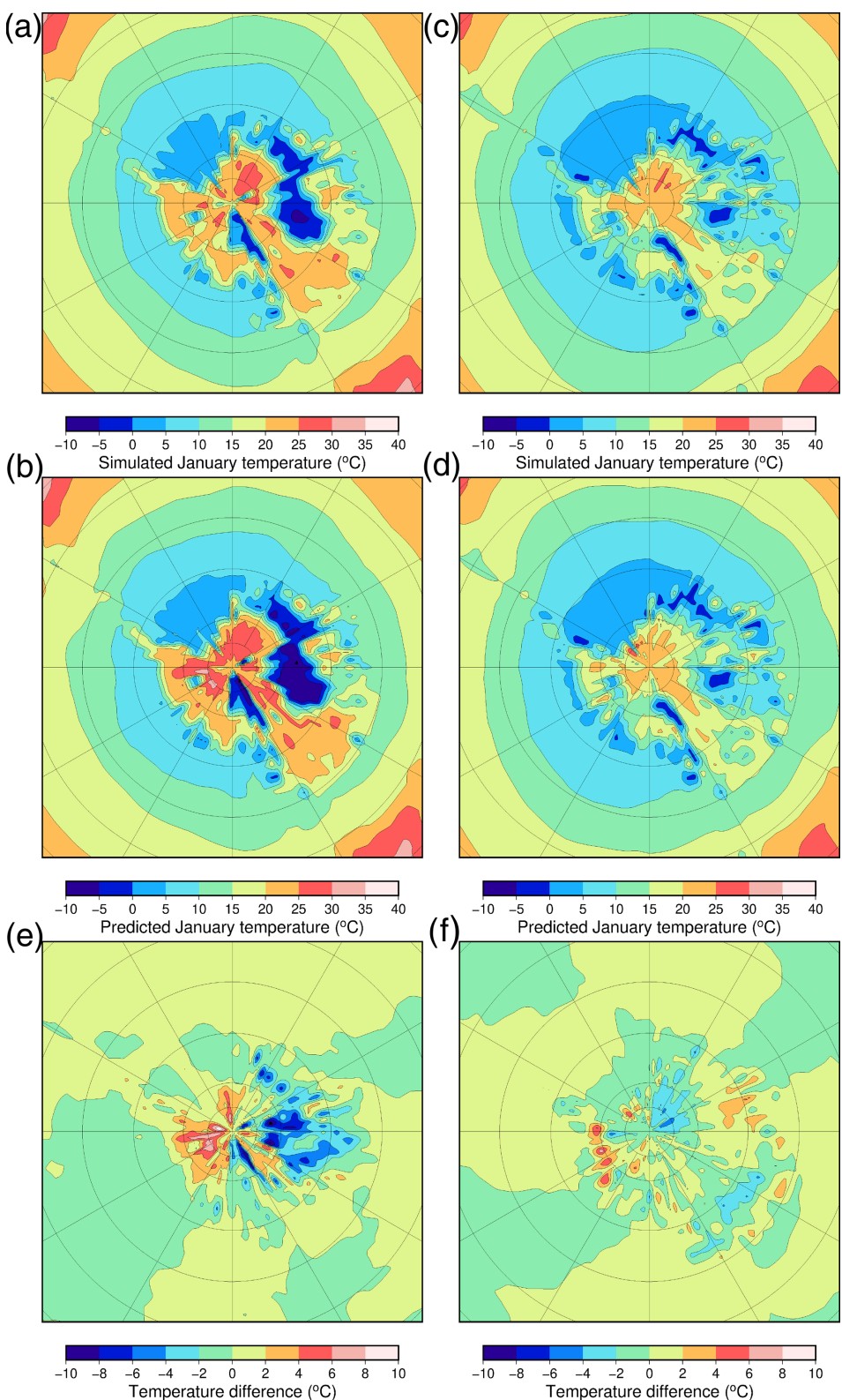

**Figure 8. (a)** Simulated and **(b)** predicted (with a leave-one-out experiment) January temperatures for the emulator showing a very poorly performing experiment (*xaemdk*) using EMULATOR_12b. **(c)** Simulated and **(d)** predicted January temperatures for a well-performing experiment (*xaemdv*) using EMULATOR_12b. Difference between the simulated and predicted temperature fields for **(e)** experiment *xaemdk* and **(f)** experiment *xaemdv*.

**Table 1.** The mean percentage of grid boxes predicted within 1 and 2 standard deviations for the four different emulators calibrated with a different ice sheet parameter. The values in bold are closest to the theoretical $1\sigma$ of 68.3 % and $2\sigma$ of 95.5 %.

| | EMULATOR_8 | | EMULATOR_12a | | EMULATOR_12b | | EMULATOR_20 | |
|---|---|---|---|---|---|---|---|---|
| | $1\sigma$ | $2\sigma$ | $1\sigma$ | $2\sigma$ | $1\sigma$ | $2\sigma$ | $1\sigma$ | $2\sigma$ |
| Ice volume | 78.8 | 97.4 | **71.2** | **94.5** | 78.4 | 97.0 | 77.3 | 96.9 |
| Ice area | 77.9 | 97.2 | 78.7 | 97.2 | 80.2 | 97.4 | 78.7 | 97.3 |
| Ice volume + ice area | 77.8 | 99.8 | 74.9 | **96.3** | 76.0 | 96.7 | 78.4 | 98.1 |

in the standard experiments), the ice sheet model passes the ice sheet parameter (the actual ice volume, ice area, or both ice volume and ice area) to the emulator and the emulator provides temperatures and precipitation as a function of the orbital parameters, $CO_2$ forcing and the ice sheet parameter. In our simulations, temperature and precipitation are interpolated to the ice sheet model grid using a bilinear interpolation scheme, in order to have smooth climatic fields. In the standard experiments, a constant lapse rate correction of $5\,°C\,km^{-1}$ is applied between HadSM3 and AISMPALEO. The climatic information is lapse rate corrected for the nearest input ice sheet geometry in terms of ice volume or ice area.

## 3 Sensitivity of the ice sheet evolution to the model set-up

In this section, the sensitivity of the ice sheet evolution is tested for the different emulators. The influence of the different number of ice sheet geometries in the model design is investigated in combination with how the ice sheet parameter is defined. In addition, the sensitivity of the ice sheet evolution is tested regarding the coupling time and the lapse rate adjustment between the coarse climate model and the much finer ice sheet model. The performance of the four different emulators is assessed. The ice sheet evolution is forced over a 3 Myr time period with the real orbital forcing from 38 to 35 Ma (Laskar et al., 2004) and $CO_2$ scenarios assuming a linear decrease in concentrations from around 980 to 720 ppmv.

### 3.1 Sensitivity to the definition of the ice sheet parameter

The ice sheet parameter represents the shape and area of the ice sheets. In previous studies (Araya-Melo et al., 2015; Lord et al., 2017), it has been defined by indexing the different ice sheet geometries that have been used to generate the simulation outputs. However, there are several other options as to how to define the ice sheet parameter. Here, it is proposed to define the ice sheet parameter by quantifying the ice sheet volume, the ice sheet area or a vector combining both aspects. This is done for the four different experiment designs. The ice sheet area and ice sheet volume are good parameters

to define the ice sheet's influence on climate because the first parameter affects the local albedo and the latter has an influence on the elevation and hence on the temperatures through adiabatic cooling. In case the ice sheet parameter is defined by both the ice sheet volume and the ice sheet area, both variables are calculated after each iteration of the ice sheet model.

Figure 9a shows the ice sheet evolution for the four emulators calibrated on ice volume. EMULATOR_12a, EMULATOR_12b and EMULATOR_20 show the transition towards a continental scale ice sheet in a very narrow $CO_2$ interval of 845 to 875 ppmv. On the other hand, EMULATOR_8 does not seem to show any sensitivity to the $CO_2$ forcing during the 3 Myr simulation (and also not on a longer timescale). The reason is that the prescribed ice sheets are separated too much in the initial climate model runs. Because of the large difference in albedo between ice and tundra, the prescribed ice sheets create a sharp boundary at the ice sheet margin that is visible in the temperature field. If insufficient prescribed ice sheet geometries are used, the ice sheet does not grow enough to see the temperature regime obtained when HadSM3 is run with the next input ice sheet geometry. Consequently, the emulated temperatures remain too high at the ice sheet margin. It appears that the threshold on the number of needed ice sheet geometries is somewhere between 8 and 12. Using 20 input ice sheet geometries does not change the model performance much.

Another option is to calibrate the emulator based on the ice area, which is more directly linked to the albedo. The glaciation threshold for EMULATOR_12a and EMULATOR_20 shows a very similar sensitivity to $CO_2$ of about 860 ppmv (Fig. 9b). EMULATOR_8 grows immediately to a medium-sized ice sheet and cannot grow further towards a continental-scale ice sheet for the reasons already quoted. EMULATOR_12b was poorly defined in terms of ice area (several ice sheet geometries had a similar area but different geometry), and the ice sheet grows immediately towards a continental scale. Therefore, in addition to having sufficient ice sheet geometries, a good spacing for the ice sheet parameter is another requirement to use an emulator for coupled ice sheet–climate simulations.

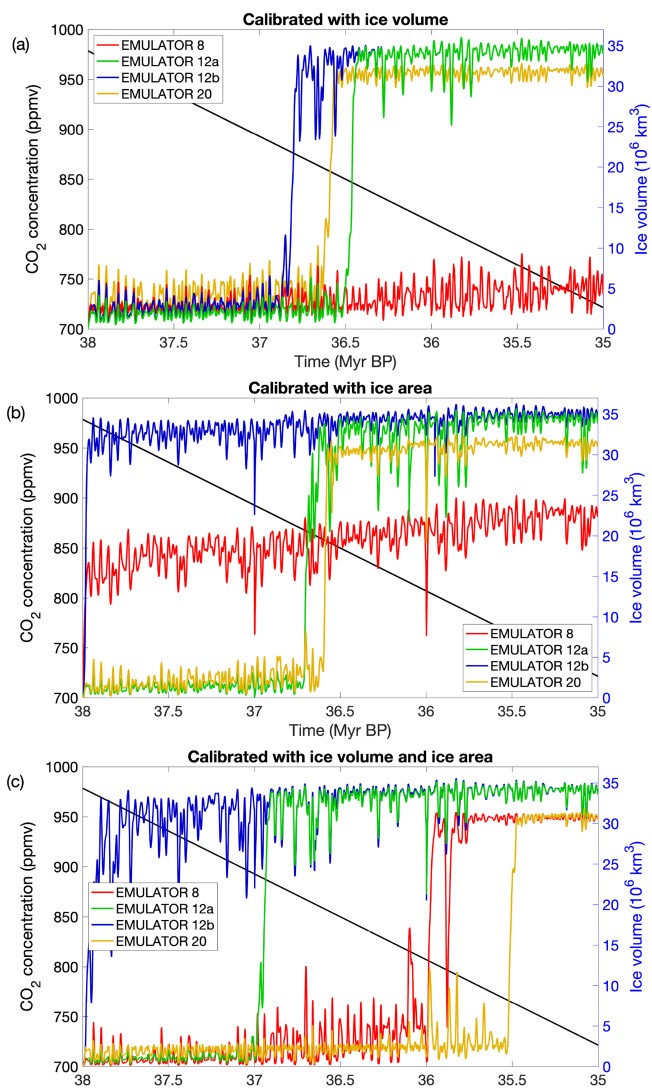

**Figure 9.** Ice sheet evolution during a 3 Myr period forced by the orbital parameters from Laskar et al. (2004) and linearly declining $CO_2$ concentrations from $\sim 980$ to $\sim 720$ ppm. Ice sheet evolution for the four different emulators calibrated based on (**a**) ice volume, (**b**) ice area, and (**c**) both ice volume and ice area.

The simulated temperatures during the first 100 kyr of the simulations are visualized during a strong insolation maximum after 47 kyr and a strong insolation minimum after 60 kyr (Fig. 10) for all four emulators calibrated on ice volume. The corresponding ice sheet sizes are given in Fig. A5 (Appendix). The temperature patterns are very similar, especially above the tundra regions. The main difference in simulated temperatures between the different emulators is caused by the size of the ice sheet, which in turn is determined by the number and spacing of ice sheets used for the calibration.

When the coupling is based on both the ice volume and the ice area, the transition to a fully glaciated climate gives very distinct results for each of the emulators (Fig. 9c). EMULA-

TOR_12a shows the transition to a continental-scale glaciation for a similar $CO_2$ threshold than for the emulators tuned on ice volume or ice area of around 890 ppmv. EMULATOR_12b shows the transition to a fully glaciated continent immediately when the simulations start because of the poor definition on ice area. Remarkably, the transition to a fully glaciated continent for EMULATOR_20 happens for a much lower $CO_2$ threshold of 765 ppmv. However, the reliability of EMULATOR_20 calibrated both on ice area and ice volume is lower than the reliability of the emulator on either ice volume or ice area (see Sect. 2.3), even though the variance is smaller when additional information on the ice sheet parameter is added. The poor emulation originates from calibrating the emulator based on six variables, while only five input forcing parameters are actually reasonably independent. The additional information on the ice sheet parameter is strongly correlated in most cases (though not everywhere) because the spread between ice volume and ice area is not equal. This is visualized in Fig. 11 where three different schematic ice sheet geometries are shown with their respective ice volume and ice area (normalized). For the second ice sheet geometry, the ice area increases by 0.8 units, while the ice volume increases by 0.2 units. In contrast, the next ice sheet geometry is defined by an ice volume increase of 0.8 units and an ice area increase of 0.2 units. In EMULATOR_20, the smallest prescribed ice sheet geometries have a significant increase in ice area, while the ice volume increase is relatively small. On the other hand, the largest prescribed ice sheet geometries have a much larger increase in ice volume than ice area (Fig. 4). For EMULATOR_12a, the relative increase in ice area and ice volume is more equal, and therefore the reliability and the performance of the emulator is better. EMULATOR_8 grows to a fully glaciated continent for a $CO_2$ threshold around 810 ppmv. As mentioned earlier, the lack of sufficient ice sheet geometries is also visible here with complete growth and decline close to the glaciation threshold.

The difference in ice sheet area and ice sheet volume evolution during the build-up of an ice sheet is further visualized in Fig. 12. When the ice sheet starts growing, both the ice sheet area and the ice volume increase. However, the timing between ice sheet volume growth and ice sheet area growth are not fully synchronous. When the climate is cool during an austral summer insolation minimum, the area where the mass balance is positive increases, and first the area where ice is accumulating increases. It takes more time for the ice volume to adjust, and by the time the ice volume reaches a maximum, the ice area starts decreasing again during an austral summer insolation maximum. This can be seen in Fig. 12c, where the first snapshot of the ice sheet geometry shows a relatively large area with ice, but ice volume at that moment is negligible. The ice area will start decreasing towards an insolation maximum, while the ice volume is still growing in response to the large area with a positive mass balance. The delay of the ice volume response compared to ice area is even

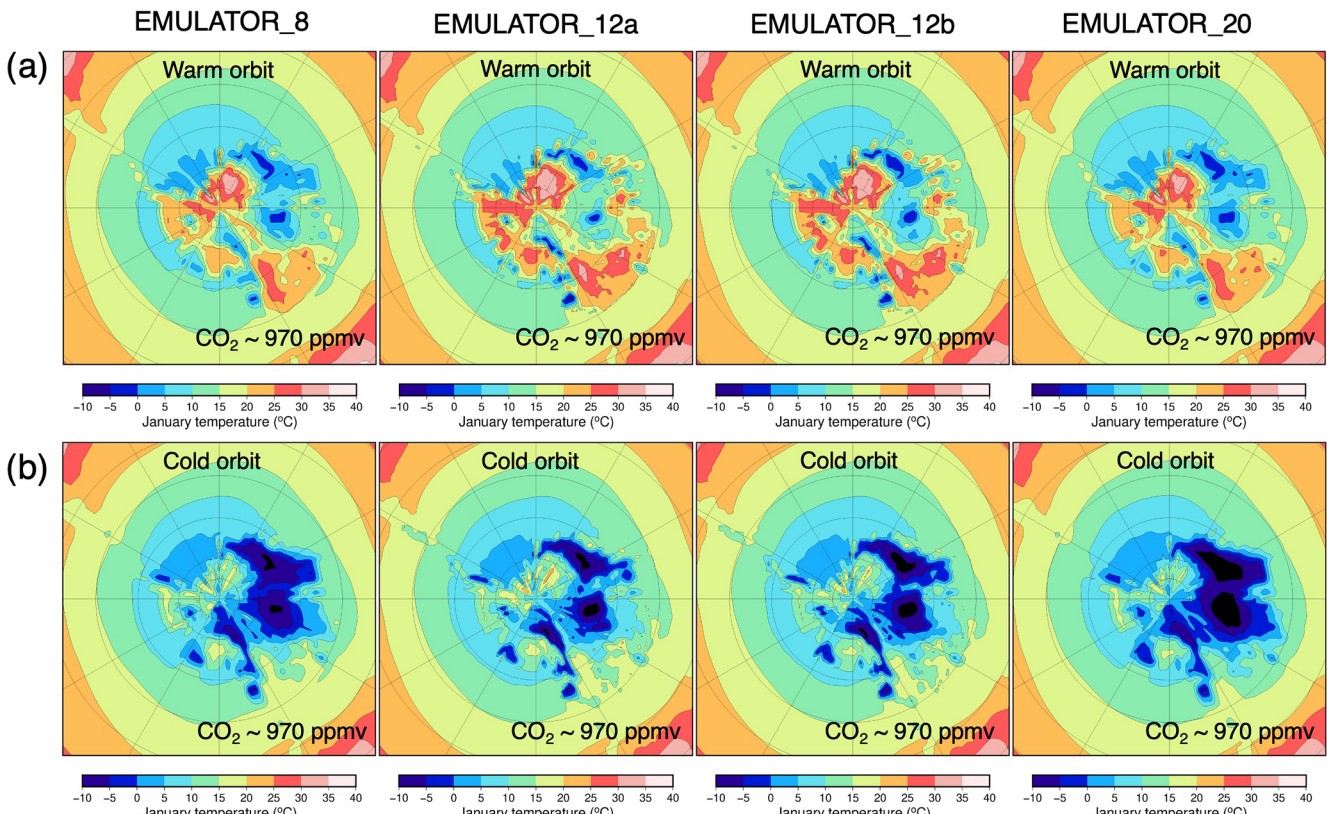

**Figure 10.** Simulated January temperature (°C) with the different emulators starting from an ice-free continent at 38 Ma after **(a)** 48 kyr (the first high-insolation maximum) and after **(b)** 60 kyr (the first high-insolation minimum) for the ice sheet parameter defined as ice volume.

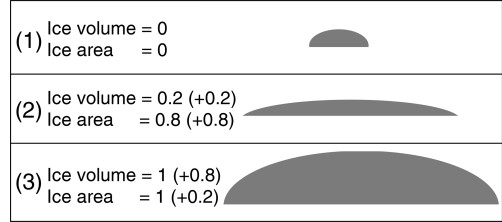

**Figure 11.** Three schematic ice sheet geometries with corresponding ice sheet volume and ice sheet area (normalized units).

more clear when an ice sheet is growing from bare bedrock to a continental-scale ice sheet(Fig. 12a and b).

These remarks suggest a possible improvement that would consist of a better experiment design with ice sheets spanning
5 a 2D space more optimally. On the other hand, the difference in relative magnitude of ice volume and ice area during the build-up of an ice sheet also suggests that it is not easy to create an optimal set of variables for the model design.

## 3.2 Sensitivity to the coupling time between AISMPALEO and the emulator

The coupling time in the first set of experiments is 1000 years. The sensitivity of the ice sheet evolution to the coupling time is tested by applying five different coupling times, ranging from 10 to 2000 years. The smallest time step
15 is of the same order of magnitude as the time step used in the ice sheet model. In this way, it can be regarded as an example of a direct coupling between the climatic component and the ice sheet model. Fig. 13 shows the ice sheet evolution during one precession cycle for the five coupling times. The climatic information for the largest time step of 2000 years
20 is updated 11 times during this interval, and the ice sheet evolution clearly responds stepwise. Another observation is that the higher the coupling time step, the more delayed the ice sheet response to the forcing and the lower the amplitude of the ice sheet volume. Decreasing the coupling time
25 step results in a smoother ice volume evolution. The differences between a coupling time of 500, 250 and 10 years become very small, suggesting that the solution converges. To make a compromise between model efficiency and model accuracy, we opted for performing the multi-million-year sen-

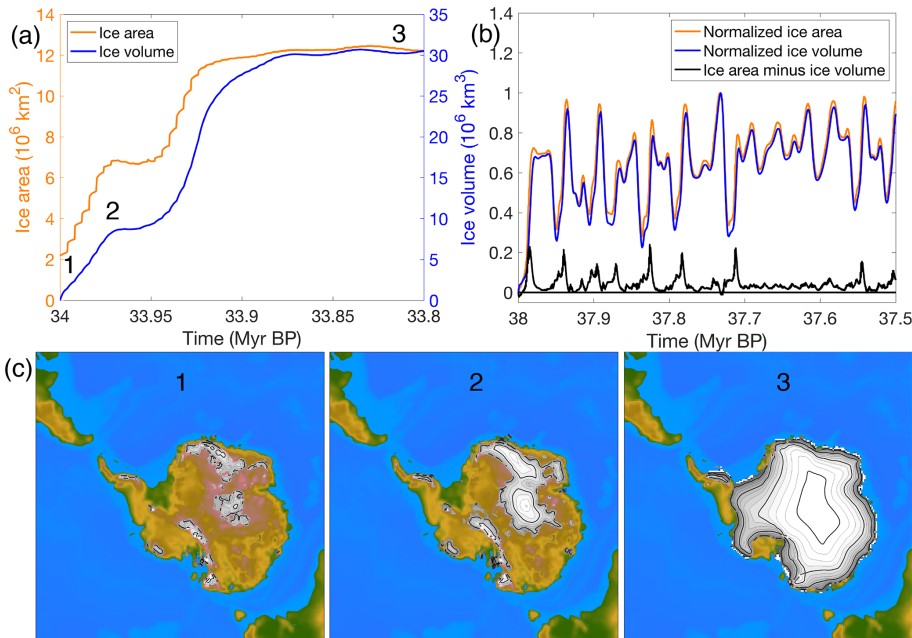

**Figure 12. (a)** Ice sheet area and ice sheet volume for an ice sheet that grows rapidly to a continental-scale ice sheet. **(b)** Normalized ice area, ice volume and the difference between both for an ice sheet that grows and melts in response to the orbital forcing. **(c)** Ice sheet geometry during three snapshots for the run when the ice sheet grows to a continental scale. The numbers in **(a)** and **(c)** indicate the time at which the snapshots are taken.

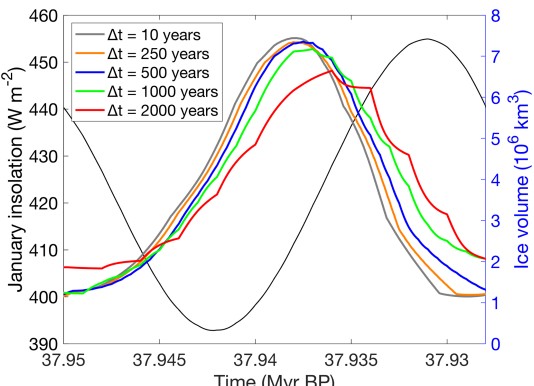

**Figure 13.** Illustration of the ice sheet volume evolution for different coupling time steps during a precession cycle ($\sim 23\,000$ years) at the beginning of the simulations. The mean January insolation is given by the thin black line.

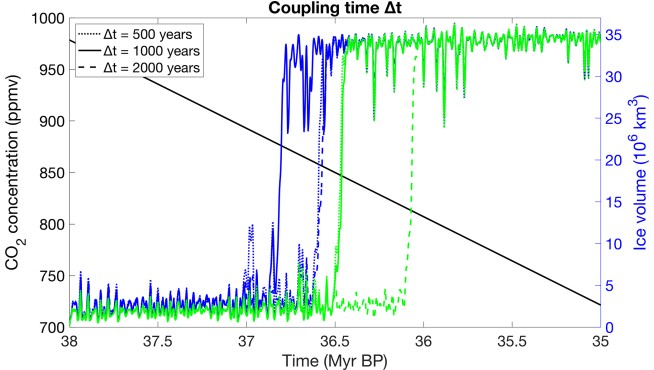

**Figure 14.** Sensitivity of the ice sheet evolution to the coupling time using the emulator calibrated with ice volume for EMULATOR_12a (green) and EMULATOR_12b (blue).

sitivity simulations with a coupling time step of 500 years as a lower limit and 2000 years as an upper limit.

The coupling time is doubled and halved to respectively 2000 and 500 years to test its influence on the ice sheet evolution for EMULATOR_12a and EMULATOR_12b calibrated with ice volume as the ice sheet parameter (Fig. 14). Halving the coupling time step increases the computational time with about 40 %, while doubling the coupling time decreases the computational time with the same percentage. When the coupling time decreases, the ice sheet volume has a larger

amplitude and is slightly more sensitive to changes in insolation.

For EMULATOR_12a, the glaciation threshold is more sensitive to $CO_2$ changes when the coupling time is decreased. The difference in glaciation threshold between a coupling time step of 500 and 1000 years is negligible, but the difference with a coupling time step of 2000 years is about 30 ppmv. The continental-scale glaciation for EMULATOR_12b with a coupling time step of 1000 years occurs for lower $CO_2$ values than for a coupling time step of 500 and 2000 years due to the complex interaction between ice

sheet response and forcing. The shorter the coupling time, the more sensitive the ice sheet is to the forcing. For a coupling time step of 1000 years, the ice sheet grows more than for a coupling time step of 2000 years, but does not decline as much as for a coupling time step of 500 years and therefore grows quicker to the fully glaciated state.

Comparing the glaciation threshold with respect to the $CO_2$ forcing between the different coupling time steps, there is a decreasing difference between EMULATOR_12a and EMULATOR_12b for a decreasing coupling time step. The difference in glaciation threshold is about 40 ppmv for a coupling time step of 2000 years, 30 ppmv for a coupling time step of 1000 years and 10 ppmv for a coupling time step of 500 years, suggesting that the solution converges for decreasing coupling time steps.

### 3.3 Sensitivity to the lapse rate adjustment between HadSM3 and AISMPALEO

The lapse rate is the change in temperature with elevation, and its value is highly dependent on the moisture in the air. Since the climate model is relatively coarse compared to the ice sheet model, we need to apply a lapse rate correction to the elevation difference between the climate model and ice sheet model. The lapse rate is, in reality, both temporarily and spatially variable, and here we test different model choices. One experiment includes temporal variations in the lapse rate correction, and another experiment includes both spatial and temporal variations. The temporally variable lapse rate is calculated as the average near-surface lapse rate for all grid points on the Antarctic continent for each month (Figs. 15 and A1). The spatially variable lapse rate is included by calculating the local near-surface lapse rate $(dT/dZ)$ simulated by HadSM3 for the four adjacent ice-covered grid points for each month. The average monthly lapse rate varies roughly between the wet adiabatic lapse rate during summer (December–January) and the dry adiabatic lapse rate in winter (June–July). These values are higher than the constant lapse rate that was applied in the standard experiments, and therefore the lapse-rate-corrected temperatures are lower for a growing ice sheet. The resulting ice sheet evolution shows a more stepwise change towards full glaciation (Fig. 16). The larger temperature difference between the climate model and the ice sheet model makes it harder for the ice sheet to grow further until a threshold is reached and a large area is cold enough for snow accumulation.

### 4 Uncertainty analysis with EMULATOR_12b calibrated on ice volume

The additional value of the use of an emulator for coupled ice sheet–climate simulations is that the mean climate predictions come with the estimate of its variance and that two different predictions have a covariance. Here, the uncertainties caused by the emulator variance are explored in order to sample climate trajectories. The covariance between output points given by the emulator is used to update the mean and variance of a climate prediction at a given iteration (e.g. time iteration $i$) of the ice sheet model, given the climate used at the previous point iteration $(i-1)$. It is then possible to sample the updated distribution. This provides a climate sample at iteration $i$, and the procedure continues to obtain climate samples at iteration $i+1$ and so forth. The process yields a sample climate trajectory. Strictly speaking, the emulated climate at iteration $i$ is correlated with the outputs at iteration $i-1, i-2$, etc., and all of them should be used to update the mean and variance at iteration $i$. However, since the Gaussian process emulator has an exponentially decaying correlation function that is short ranged (in contrast to a power law), it is expected that the covariance structure of emulated climate trajectories that is associated with the emulator variance is effectively captured by the first-order autocorrelation.

Now that the general principle is explained, more mathematical details about the procedure are provided with specific attention to the fact that a PCA emulator is used. At any time step $i$, the current temperature is estimated on the basis of principal components. To this end, the emulator mean is computed for each PC score, given the orbital parameters, the $CO_2$ concentration and the ice level at the current time step $m(\boldsymbol{x}_i)$ and at the previous time step $m(\boldsymbol{x}_{i-1})$, along with the computed covariances associated with these elements, denoted $V(\boldsymbol{x}_i, \boldsymbol{x}_i)$, $V(\boldsymbol{x}_{i-1}, \boldsymbol{x}_i)$ and $V(\boldsymbol{x}_{i-1}, \boldsymbol{x}_{i-1})$. The mean and covariance at time step $i$ are then updated given the scores for the corresponding principal component $T_{i-1}$ TS5, which was effectively applied at time step $i-1$. The PC score at time step $i$, which will finally be applied to the ice sheet model, is drawn from this distribution. The procedure is repeated for each PC score. The temperature field reconstructed from these PC scores is further perturbed by a random field with variance equal to the residual variance not captured by the principal components. This approach provides us with a random draw of the temperature field consistent with the information provided by the PCA emulator (Eqs. 6–8).

$$m(\boldsymbol{x}_i) \vee T_{i-1} = m(\boldsymbol{x}_i) + \frac{V(\boldsymbol{x}_i, \boldsymbol{x}_{i-1})}{V(\boldsymbol{x}_{i-1}, \boldsymbol{x}_{i-1})} \\ \times (T_{i-1} - m(\boldsymbol{x}_{i-1})) \tag{6}$$

$$\sigma^2(\boldsymbol{x}_i) \vee T_{i-1} = V(\boldsymbol{x}_i, \boldsymbol{x}_i) \\ \times \left(1 - \left(\frac{V(\boldsymbol{x}_i, \boldsymbol{x}_{i-1})^2}{V(\boldsymbol{x}_i, \boldsymbol{x}_i) V(\boldsymbol{x}_{i-1}, \boldsymbol{x}_{i-1})}\right)\right) \tag{7}$$

$$T_i \sim N\left(m(\boldsymbol{x}_i) \vee T_{i-1}, \sigma^2(\boldsymbol{x}_i) \vee T_{i-1}\right) \tag{8}$$

The uncertainty of the emulator is explored by performing 50 Monte Carlo simulations including the variance of EMULATOR_12b calibrated with ice volume. Since temperature and precipitation are emulated separately for each month, a

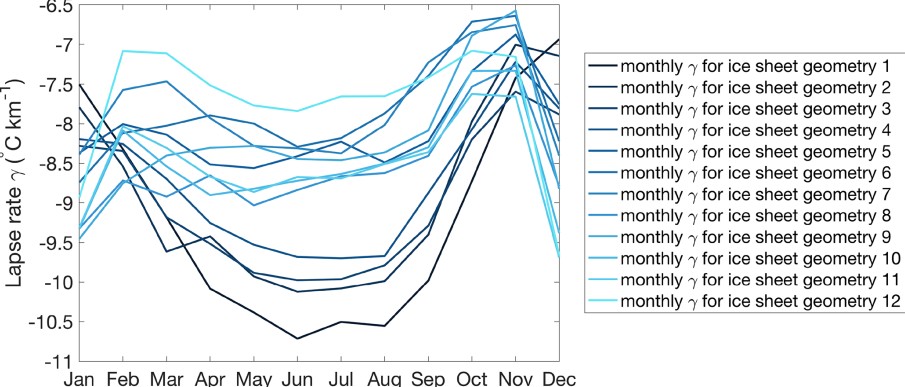

**Figure 15.** Average monthly near-surface lapse rate in HadSM3 over the Antarctic continent for each of the 12 ice sheet geometries. Ice sheet geometry 1 is the largest ice sheet and ice sheet geometry 12 is the smallest ice sheet.

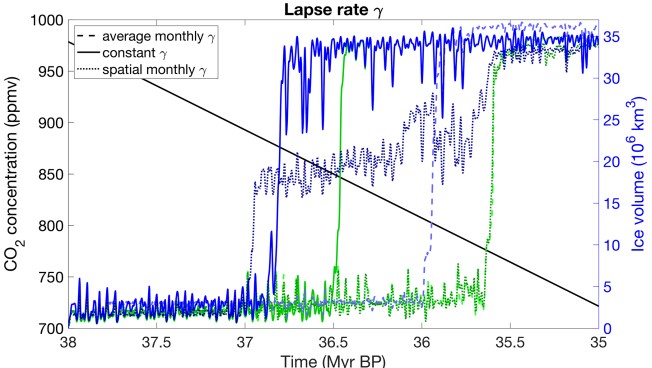

**Figure 16.** Sensitivity of the ice sheet evolution to the application of the lapse rate correction using the emulator calibrated with ice volume for EMULATOR_12a (green) and EMULATOR 12b (blue).

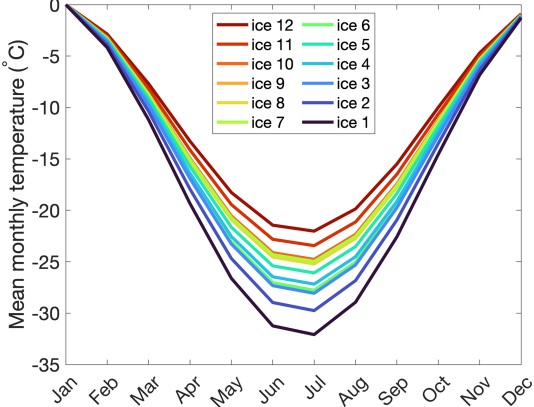

**Figure 17.** Annual cycle for each of the 12 input ice sheet geometries with respect to January temperatures.

choice had to be made as to which parameter is most decisive for ice sheet growth. It appears that summer temperature has a main control on the evolution of the ice sheet over time. Therefore, the uncertainties in the emulated January temperature are explored and emulated precipitation in a similar manner as the previous experiments. The temperatures of the other months are reconstructed based on the annual cycle of the 100 input experiments. First, the mean temperature for each month and for each input ice sheet geometry is calculated. Following this, the annual cycle with respect to the January temperatures is calculated for each input ice sheet geometry (Fig. 17). The mean difference between each month and the emulated January temperature is applied to calculate the temperature of the other months. By applying a constant temperature anomaly compared to the January temperature, the temperature of the ice-free regions in January is overestimated for the June temperatures when these regions are snow covered. This has a minor influence on the results because ice melt does not occur during the austral winter months anyway. Therefore, the emulator is almost identical to EMULATOR_12b, except that covariances are used to sample trajectories around the mean. If the variances are assumed to be zero at all time steps, the mean trajectory already presented in Fig. 9a is approximated (slight difference due to the application of the annual cycle to the January temperature).

The resulting ice sheet evolution over time is shown in Fig. 18 for a coupling time of 500 years and a coupling time of 2000 years. The original simulations are shown by the blue curve, and the approximation by emulating only January temperatures and applying a constant correction based on the annual cycle is represented by the green curve. Generally, the curves look very similar, but the simulations with emulating only the January temperatures slightly underestimate the ice sheet volume compared to the original run and result in a glaciation threshold occurring for lower $CO_2$ values than for the original run. This is the result of applying a constant temperature correction over the entire continent with respect to the January temperature. The actual austral

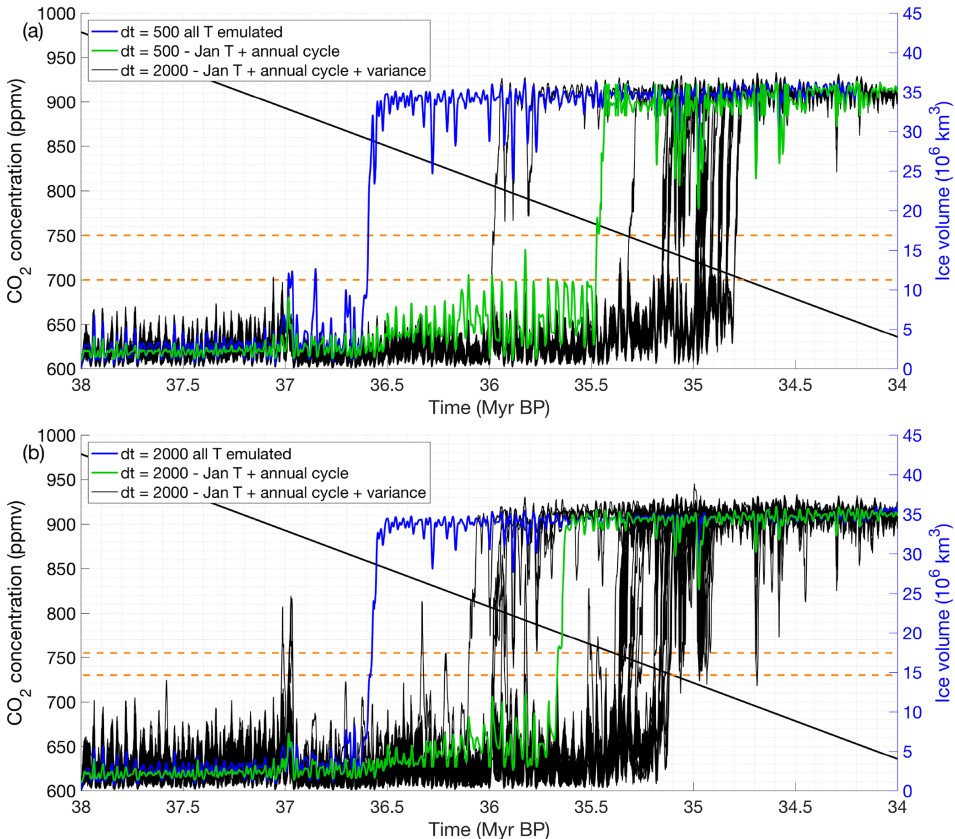

**Figure 18.** Monte Carlo simulations showing the ice sheet evolution for a coupling step of **(a)** 500 years and **(b)** 2000 years. The simulation ran with the emulated January temperatures and the annual cycle is shown by the green line, and the original run is shown by the blue line. Horizontal dashed orange lines indicate the interval at which most experiments reach the glaciation threshold.

autumn, winter and spring temperatures are colder in the ice-free regions due to the effect of snowfall on the albedo.

The black curves represent 50 Monte Carlo simulations where the variance of the January temperature is added to the mean predictions. Both simulations, with a coupling time step of 500 and 2000 years, have a similar variance. The overall uncertainty in the ice sheet evolution is also comparable for both coupling time steps with a difference between the lowest and highest glaciation threshold of about 25–50 ppmv for more than 95 % of the simulations. The asymmetry in the glaciation threshold is striking for the experiments including the variance in comparison with the reference experiment (Fig. 18). Most experiments including the variance predict the glaciation threshold to happen for lower $CO_2$ values than the reference experiment based on the mean temperature prediction. The reason is that the ice sheet in these simulations loses mass during an insolation maximum for a longer time and therefore does not manage to grow above the mean prediction. Fig. 18b also shows that the ice sheet is growing (peaks are higher) and melting (the ice sheet is not that stable) faster for a coupling time of 2000 years compared to a coupling time of 500 years. The main mechanism is the slow response time of the ice sheet that has more time

to grow during an insolation minimum and more time to decay during an insolation maximum for a larger coupling time step.

## 5   Discussion

The aim of the new coupling technique CLISEMv1.0 is to create an efficient and accurate way to model ice sheet–climate interactions on timescales beyond what directly coupled models can achieve. In that sense, we build further on previous modelling attempts such as the asynchronous coupling method and the matrix method. The basic asynchronous method has the advantage that you do not need to have any prior information on the possible ice sheet geometries. A strong disadvantage is that this method does not allow for sensitivity experiments at a reasonable computing time since the whole chain of ice sheet model and GCM runs would have to be repeated. Nevertheless, this method has been very popular in paleoclimatic studies during all time periods in geological history where ice might have been present, from the Paleozoic ice houses (Horton et al., 2007; Lowry et al., 2014; Pohl et al., 2016) over the Eocene–

Oligocene transition (DeConto and Pollard, 2003), Miocene (Gasson et al., 2016) to the Quaternary ice ages (Charbit et al., 2002; Herrington and Poulsen, 2012).

With CLISEMv1.0, the forcing uncertainty is explored with preliminary GCM snapshots. The emulation of climate and precipitation is akin to kriging or geospatial interpolation, but in five- or six-dimensional space with a large number of climate model runs. It is not the same as linear interpolation, as the posterior mean includes a term that absorbs deviations from linearity. Crucially, the emulator comes with estimates of covariance, which measures the uncertainty introduced by using the emulator. Checking that this uncertainty is consistent with leave-one-out experiments is a key aspect of the emulator evaluation. A so-called "nugget" allows the introduction of variability directly explained by the model inputs, such as model internal variability (Andrianakis and Challenor, 2012), but in our design this nugget is a small numerical value. For this reason, the use of a GP emulator is also not completely equal to interpolating the raw model output from the climate model. This is in contrast to the climate matrix method, which consists of a limited number of GCM runs for the endmembers in the forcing and linearly interpolates the climatologies based on the actual ice sheet geometry (Gasson et al., 2016; Stap et al., 2017; Berends et al., 2018).

Whether the asynchronous coupling method, the matrix lookup table or the GP emulator is used to simulate ice sheet–climate interactions on (multi)-million-year timescales, a choice about the coupling time has to be made and will affect the outcome. When using the matrix lookup table or the emulator, the result will also be dependent on the choice of the model design and on the number of GCM runs. For the GP emulator, additional choices need to be made about the length scale, nugget and covariance function to use. The more complexity is added to the method, the more uncertainties might arise. On the other hand, the GP emulator provides a posterior covariance that provides an objective criterion to verify that it is well calibrated and to evaluate the introduced uncertainties.

A common problem for the emulator and the matrix lookup table method, where the ice sheet parameter is defined by a single number, is that there is no control on the regions where ice starts to grow. The problem can be addressed by describing the ice sheet location and geometry with a vector of several dimensions. In reverse, the definition of this vector and the experiment design have to provide a reasonably orthogonal experiment design in order to avoid the issues experienced in this study by attempting to calibrate the emulator both on ice volume and ice area simultaneously. Optionally, ice sheets could be described with additional variables such as shape factors that are relatively independent of the other ice sheet parameters. We leave the suggestion of creating other ice sheet variables to improve the emulator performance for future work.

To have a properly working emulator that includes dynamic ice sheets, it is crucial to have a good spacing be-

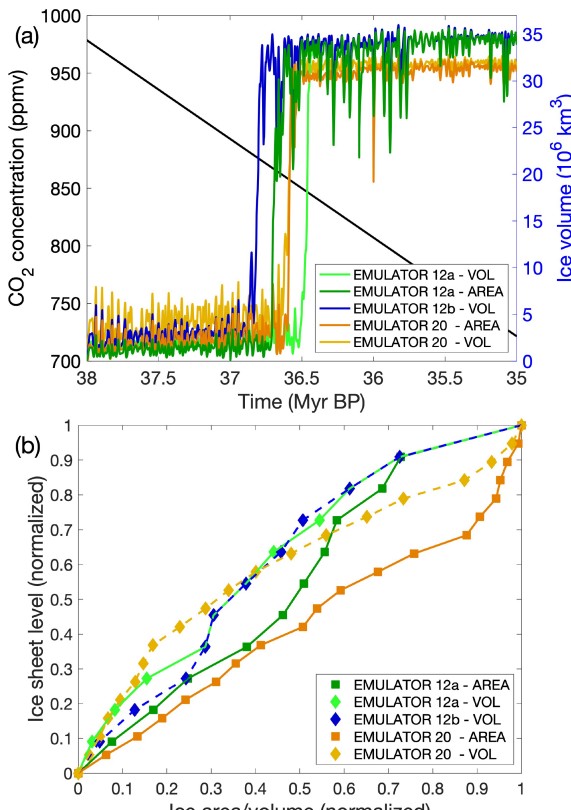

**Figure 19. (a)** Ice sheet evolution for EMULATOR_12b tuned on ice volume, EMULATOR_12a tuned on ice volume and ice area, EMULATOR_20 tuned on ice volume and ice area. **(b)** Comparison of the input ice sheet geometry spacing when looking at ice area and ice volume for EMULATOR_12a, EMULATOR_12b (only ice volume) and EMULATOR_20.

tween the input ice sheet geometries and sufficient different ice sheet geometries. Five different emulators have been shown to have a good set-up: EMULATOR_12a, EMULATOR_12b and EMULATOR_20 tuned on ice volume and EMULATOR_12a and EMULATOR_20 tuned on ice area (Fig. 19). EMULATOR_8 included too few input ice sheet geometries to make the ice sheet grow to a continental-scale ice sheet. EMULATOR_12b tuned on ice area did not produce reliable results because several input ice sheet areas had a different geometry but similar area. The glaciation threshold for each of these five emulators ranges between 845 and 875 ppmv. Overall, taking into account the radiative forcing of carbon dioxide, the differences in glaciation threshold are very small between the different ways of calibrating the emulators (see the video supplement for the ice sheet geometry evolution of EMULATOR_12a, EMULATOR_12b and EMULATOR_20 calibrated on ice volume).

It appears that the more prescribed ice sheet geometries are used in the emulator, the larger the amplitude of ice sheet growth during a cold orbital configuration. However, this has no impact on the $CO_2$ threshold to continental-scale glacia-

tion. The maximum ice sheet extent is also dependent on the largest predefined ice sheet geometry. EMULATOR_12a and EMULATOR_12b had a smaller largest predefined ice sheet geometry than EMULATOR_20. The extrapolation of the emulated climatic variables led to a climatic state that was too cold, allowing for a larger ice sheet than EMULATOR_20 that was calibrated on a larger range of prescribed ice sheet geometries.

Our simulations show that using a coupling time step of 500 years instead of 2000 years results in a quicker ice sheet response to the forcing. Such small coupling time steps are not common for multi-million-year simulations with three-dimensional ice sheet models coupled to a climate model. Gasson et al. (2016) used an asynchronous coupling method to simulate the ice sheet evolution during the Miocene with a coupling step of 2000 years, while Stap et al. (2017) used a coupling time step of 500 years for a one-dimensional ice sheet model forced by a climate model. The value for the lapse rate correction clearly also has an influence on the ice sheet evolution over time, and the emulator allows a number of sensitivity experiments. We have attempted to include a more realistic lapse rate that follows the seasonal and spatial variations. In HadSM3, the monthly average lapse rate over the Antarctic ice sheet ranges between $7\,°C\,km^{-1}$ in January (summer) to $10.5\,°C\,km^{-1}$ in July (winter). The lapse rate over the Greenland ice sheet during the Last Glacial Maximum had a similar range from $\sim 5.5\,°C\,km^{-1}$ during summer to $9.5\,°C\,km^{-1}$ during winter (Erokhina et al., 2017). This near-surface lapse rate is influenced by atmospheric boundary processes, the surface type (snow, tundra) and the atmospheric circulation (Kageyama et al., 2005).

## 6    Conclusions

In this study, the computationally efficient coupler CLISEMv1.0 that provides climatic fields for simulating ice sheet–climate interactions on a multi-million-year timescale has been described together with its sensitivity to the implementation and an uncertainty analysis. CLISEMv1.0 estimates the climate as a function of the orbital parameters, the $CO_2$ forcing and the ice sheet parameter, where each forcing is defined by a single number. The ice sheet parameter is either defined by the ice sheet area, the ice sheet volume, or both the ice sheet area and ice sheet volume.

A set of different emulators was constructed to investigate the influence of the number of prescribed ice sheet geometries in the model design on the coupled ice sheet–climate simulations. The number of precursor ice sheet geometries has a large effect on the ice sheet sensitivity to $CO_2$ and orbital forcing because of its large climatic imprint caused by the high albedo of ice. In addition, the spread of the ice sheet geometries has been shown to have a significant impact on the performance of the coupled ice sheet–climate simulations

and has a larger effect than the definition of the ice sheet parameter. When there is an equal spread between the ice sheet area and the ice sheet volume of the input ice sheet geometries, the threshold to continental-scale glaciation occurs in a very narrow $CO_2$ window of $860 \pm 15$ ppmv.

Once the emulator is well calibrated, the emulator–ice sheet coupling method is very suitable to use for performing ice sheet–climate simulations on a multi-million-year timescale and to use for sensitivity tests. Here we tested the sensitivity of the ice sheet evolution to the coupling time and to the lapse rate adjustment. Our results indicated that the glaciation threshold to the $CO_2$ forcing converges for a decreasing time step. In addition, shortening the coupling time slightly increases the sensitivity to $CO_2$ forcing. The shorter the coupling time, the larger the ice sheet grows during an insolation minimum and the more the ice sheets shrinks during an insolation maximum. This might have large consequences for paleoclimatic studies implementing asynchronous coupling techniques, where the coupling time is usually on the order of several millennia. The elevation differences between coarse climate models and high-resolution ice sheet models are usually corrected for by applying a constant lapse rate correction. The value of this lapse rate correction has a larger effect than the coupling time, and we propose taking the real lapse rate correction that is observed in the climate model output.

The emulator–ice sheet coupling method is applied here for idealized $CO_2$ scenarios for the time period between 38 and 35 Ma. In these simulations, temperature and precipitation are emulated to drive the mass balance of the ice sheet model. CLISEMv1.0 is a useful tool to investigate the ice sheet evolution during all major climatic transitions of the Cenozoic, where the interaction between orbital parameters and $CO_2$ variations are thought to have played a significant role, or even to investigate the existence of pre-Cenozoic glaciations throughout the Phanerozoic.

## Appendix A

**Table A1.** Experiments with their name, orbital parameter values, $CO_2$ values, and ice level expressed in terms of ice volume (VOL) and ice area (AR) for EMULATOR_8 (EM_8), EMULATOR_12a (EM_12a), EMULATOR_12b (EM_12b) and EMULATOR_20 (EM_20).

| No. | Name | $e$ | $\tilde{\omega}$ | $\varepsilon$ | $CO_2$ | EM_8 VOL/AR [$10^7$ km$^3$ km$^{-2}$] | EM_12a VOL/AR [$10^7$ km$^3$ km$^{-2}$] | EM_12b VOL/AR [$10^7$ km$^3$ km$^{-2}$] | EM_20 VOL/AR [$10^7$ km$^3$ km$^{-2}$] |
|---|---|---|---|---|---|---|---|---|---|
| 1 | xaemaa | 0.0492 | 194.7 | 22.14 | 552.1 | 3.29/13.26 | 2.35/10.98 | 2.35/10.98 | 3.29/13.26 |
| 2 | xaemab | 0.0523 | 237.0 | 23.36 | 709.1 | 3.29/13.26 | 2.35/10.98 | 2.35/10.98 | 3.29/13.26 |
| 3 | xaemac | 0.0090 | 198.1 | 23.68 | 1143.9 | 3.29/13.26 | 2.35/10.98 | 2.35/10.98 | 3.29/13.26 |
| 4 | xaemad | 0.0610 | 276.3 | 22.07 | 804.3 | 3.29/13.26 | 2.35/10.98 | 2.35/10.98 | 3.29/13.26 |
| 5 | xaemae | 0.0514 | 302.4 | 23.12 | 946.1 | 3.29/13.26 | 2.35/10.98 | 2.35/10.98 | 3.29/13.26 |
| 6 | xaemaf | 0.0122 | 353.1 | 22.53 | 1140.8 | 3.29/13.26 | 2.35/10.98 | 2.35/10.98 | 3.22/13.17 |
| 7 | xaemag | 0.0425 | 19.2 | 23.22 | 679.0 | 3.29/13.26 | 2.35/10.98 | 2.35/10.98 | 3.22/13.17 |
| 8 | xaemah | 0.0264 | 168.7 | 22.47 | 562.4 | 3.29/13.26 | 2.35/10.98 | 2.35/10.98 | 3.22/13.17 |
| 9 | xaemai | 0.0273 | 245.8 | 24.17 | 727.9 | 3.29/13.26 | 1.71/8.04 | 1.71/8.04 | 3.22/13.17 |
| 10 | xaemaj | 0.0306 | 306.4 | 22.97 | 660.3 | 3.29/13.26 | 1.71/8.04 | 1.71/8.04 | 3.22/13.17 |
| 11 | xaemak | 0.0467 | 302.0 | 22.73 | 640.5 | 3.29/13.26 | 1.71/8.04 | 1.71/8.04 | 3.07/12.86 |
| 12 | xaemal | 0.0438 | 355.6 | 23.86 | 622.8 | 3.29/13.26 | 1.71/8.04 | 1.71/8.04 | 3.07/12.86 |
| 13 | xaemam | 0.0589 | 12.1 | 22.85 | 788.9 | 2.42/12.54 | 1.71/8.04 | 1.71/8.04 | 3.07/12.86 |
| 14 | xaeman | 0.0396 | 105.9 | 23.91 | 665.0 | 2.42/12.54 | 1.71/8.04 | 1.71/8.04 | 3.07/12.86 |
| 15 | xaemao | 0.0422 | 176.6 | 24.42 | 720.1 | 2.42/12.54 | 1.71/8.04 | 1.71/8.04 | 3.07/12.86 |
| 16 | xaemap | 0.0248 | 148.3 | 23.02 | 965.5 | 2.42/12.54 | 1.71/8.04 | 1.71/8.04 | 2.87/12.66 |
| 17 | xaemaq | 0.0078 | 167.7 | 22.34 | 579.9 | 2.42/12.54 | 1.71/8.04 | 1.71/8.04 | 2.87/12.66 |
| 18 | xaemar | 0.0145 | 183.1 | 23.39 | 918.5 | 2.42/12.54 | 1.44/7.59 | 1.44/7.59 | 2.87/12.66 |
| 19 | xaemas | 0.0348 | 212.5 | 22.91 | 589.2 | 2.42/12.54 | 1.44/7.59 | 1.44/7.59 | 2.87/12.66 |
| 20 | xaemat | 0.0309 | 276.1 | 23.14 | 690.0 | 2.42/12.54 | 1.44/7.59 | 1.44/7.59 | 2.87/12.66 |
| 21 | xaemau | 0.0490 | 322.6 | 24.26 | 770.8 | 2.42/12.54 | 1.44/7.59 | 1.44/7.59 | 2.42/12.54 |
| 22 | xaemav | 0.0442 | 38.7 | 23.94 | 686.0 | 2.42/12.54 | 1.44/7.59 | 1.44/7.59 | 2.42/12.54 |
| 23 | xaemaw | 0.0199 | 0.2 | 22.10 | 879.2 | 2.42/12.54 | 1.44/7.59 | 1.44/7.59 | 2.42/12.54 |
| 24 | xaemax | 0.0564 | 136.4 | 24.47 | 824.5 | 2.42/12.54 | 1.44/7.59 | 1.44/7.59 | 2.42/12.54 |
| 25 | xaemay | 0.0557 | 149.6 | 22.25 | 912.8 | 2.42/12.54 | 1.44/7.59 | 1.44/7.59 | 2.42/12.54 |
| 26 | xaemaz | 0.0499 | 230.0 | 22.28 | 1012.5 | 1.85/11.70 | 1.29/6.48 | 1.20/6.21 | 2.15/12.08 |
| 27 | xaemba | 0.0272 | 269.2 | 23.26 | 727.5 | 1.85/11.70 | 1.29/6.48 | 1.20/6.21 | 2.15/12.08 |
| 28 | xaembb | 0.0627 | 85.3 | 24.28 | 635.1 | 1.85/11.70 | 1.29/6.48 | 1.20/6.21 | 2.15/12.08 |
| 29 | xaembc | 0.0134 | 25.5 | 22.16 | 567.3 | 1.85/11.70 | 1.29/6.48 | 1.20/6.21 | 2.15/12.08 |
| 30 | xaembd | 0.0552 | 88.7 | 22.78 | 699.7 | 1.85/11.70 | 1.29/6.48 | 1.20/6.21 | 2.15/12.08 |
| 31 | xaembe | 0.0467 | 90.4 | 24.08 | 1083.2 | 1.85/11.70 | 1.29/6.48 | 1.20/6.21 | 1.85/11.70 |
| 32 | xaembf | 0.0431 | 128.6 | 23.33 | 1132.4 | 1.85/11.70 | 1.29/6.48 | 1.20/6.21 | 1.85/11.70 |
| 33 | xaembg | 0.0220 | 153.6 | 22.48 | 1039.7 | 1.85/11.70 | 1.29/6.48 | 1.20/6.21 | 1.85/11.70 |
| 34 | xaembh | 0.0161 | 247.9 | 22.18 | 1036.1 | 1.85/11.70 | 1.04/6.19 | 1.08/5.74 | 1.85/11.70 |
| 35 | xaembi | 0.0165 | 270.7 | 23.32 | 888.9 | 1.85/11.70 | 1.04/6.19 | 1.08/5.74 | 1.85/11.70 |
| 36 | xaembj | 0.0281 | 78.9 | 23.97 | 1070.3 | 1.85/11.70 | 1.04/6.19 | 1.08/5.74 | 1.59/10.21 |
| 37 | xaembk | 0.0106 | 48.8 | 23.30 | 740.6 | 1.85/11.70 | 1.04/6.19 | 1.08/5.74 | 1.59/10.21 |
| 38 | xaembl | 0.0242 | 80.5 | 22.32 | 1025.3 | 1.33/9.18 | 1.04/6.19 | 1.08/5.74 | 1.59/10.21 |
| 39 | xaembm | 0.0543 | 157.1 | 22.39 | 751.0 | 1.33/9.18 | 1.04/6.19 | 1.08/5.74 | 1.59/10.21 |
| 40 | xaembn | 0.0267 | 167.0 | 24.48 | 748.6 | 1.33/9.18 | 1.04/6.19 | 1.08/5.74 | 1.59/10.21 |
| 41 | xaembo | 0.0230 | 166.8 | 23.63 | 772.9 | 1.33/9.18 | 1.04/6.19 | 1.08/5.74 | 1.33/9.18 |
| 42 | xaembp | 0.0026 | 210.6 | 23.06 | 628.6 | 1.33/9.18 | 0.90/5.68 | 0.90/5.68 | 1.33/9.18 |
| 43 | xaembq | 0.0443 | 184.2 | 22.28 | 1103.3 | 1.33/9.18 | 0.90/5.68 | 0.90/5.68 | 1.33/9.18 |
| 44 | xaembr | 0.0396 | 198.5 | 23.61 | 841.9 | 1.33/9.18 | 0.90/5.68 | 0.90/5.68 | 1.33/9.18 |
| 45 | xaembs | 0.0523 | 254.0 | 23.23 | 673.6 | 1.33/9.18 | 0.90/5.68 | 0.90/5.68 | 1.33/9.18 |
| 46 | xaembt | 0.0610 | 271.5 | 24.04 | 646.2 | 1.33/9.18 | 0.90/5.68 | 0.90/5.68 | 1.13/8.12 |
| 47 | xaembu | 0.0428 | 282.6 | 23.72 | 574.0 | 1.33/9.18 | 0.90/5.68 | 0.90/5.68 | 1.13/8.12 |
| 48 | xaembv | 0.0445 | 0.3 | 22.01 | 1005.6 | 1.33/9.18 | 0.90/5.68 | 0.90/5.68 | 1.13/8.12 |
| 49 | xaembw | 0.0398 | 3.3 | 24.06 | 782.6 | 1.33/9.18 | 0.90/5.68 | 0.90/5.68 | 1.13/8.12 |
| 50 | xaembx | 0.0561 | 93.7 | 22.21 | 1112.0 | 0.77/7.05 | 0.90/5.68 | 0.90/5.68 | 1.13/8.12 |

| No. | Name | $e$ | $\tilde{\omega}$ | $\varepsilon$ | $CO_2$ | EM_8 VOL/AR $[10^7\ km^3\ km^{-2}]$ | EM_12a VOL/AR $[10^7\ km^3\ km^{-2}]$ | EM_12b VOL/AR $[10^7\ km^3\ km^{-2}]$ | EM_20 VOL/AR $[10^7\ km^3\ km^{-2}]$ |
|---|---|---|---|---|---|---|---|---|---|
| 51 | xaemby | 0.0310 | 304.5 | 22.89 | 594.5 | 0.77/7.05 | 0.72/5.17 | 0.72/5.17 | 0.96/7.45 |
| 52 | xaembz | 0.0479 | 342.0 | 23.48 | 1121.1 | 0.77/7.05 | 0.72/5.17 | 0.72/5.17 | 0.96/7.45 |
| 53 | xaemca | 0.0270 | 359.6 | 23.80 | 556.1 | 0.77/7.05 | 0.72/5.17 | 0.72/5.17 | 0.96/7.45 |
| 54 | xaemcb | 0.0163 | 12.8 | 22.94 | 736.9 | 0.77/7.05 | 0.72/5.17 | 0.72/5.17 | 0.96/7.45 |
| 55 | xaemcc | 0.0253 | 42.1 | 23.55 | 812.4 | 0.77/7.05 | 0.72/5.17 | 0.72/5.17 | 0.96/7.45 |
| 56 | xaemcd | 0.0346 | 39.8 | 24.21 | 884.2 | 0.77/7.05 | 0.72/5.17 | 0.72/5.17 | 0.77/7.05 |
| 57 | xaemce | 0.0579 | 120.1 | 23.88 | 982.1 | 0.77/7.05 | 0.72/5.17 | 0.72/5.17 | 0.77/7.05 |
| 58 | xaemcf | 0.0591 | 161.4 | 24.33 | 1097.2 | 0.77/7.05 | 0.72/5.17 | 0.72/5.17 | 0.77/7.05 |
| 59 | xaemcg | 0.0481 | 184.6 | 23.67 | 1050.7 | 0.77/7.05 | 0.68/4.29 | 0.68/4.29 | 0.77/7.05 |
| 60 | xaemch | 0.0476 | 184.1 | 24.12 | 956.1 | 0.77/7.05 | 0.68/4.29 | 0.68/4.29 | 0.77/7.05 |
| 61 | xaemci | 0.0237 | 210.0 | 24.32 | 1087.7 | 0.77/7.05 | 0.68/4.29 | 0.68/4.29 | 0.57/5.85 |
| 62 | xaemcj | 0.0524 | 259.7 | 23.55 | 854.8 | 0.77/7.05 | 0.68/4.29 | 0.68/4.29 | 0.57/5.85 |
| 63 | xaemck | 0.0107 | 294.2 | 22.76 | 813.3 | 0.50/5.14 | 0.68/4.29 | 0.68/4.29 | 0.57/5.85 |
| 64 | xaemcl | 0.0389 | 0.9 | 22.68 | 614.4 | 0.50/5.14 | 0.68/4.29 | 0.68/4.29 | 0.57/5.85 |
| 65 | xaemcm | 0.0289 | 19.3 | 22.52 | 801.0 | 0.50/5.14 | 0.68/4.29 | 0.68/4.29 | 0.57/5.85 |
| 66 | xaemcn | 0.0279 | 96.1 | 22.71 | 604.7 | 0.50/5.14 | 0.68/4.29 | 0.68/4.29 | 0.50/5.14 |
| 67 | xaemco | 0.0407 | 95.4 | 22.03 | 650.0 | 0.50/5.14 | 0.68/4.29 | 0.68/4.29 | 0.50/5.14 |
| 68 | xaemcp | 0.0226 | 277.6 | 22.57 | 666.3 | 0.50/5.14 | 0.37/2.85 | 0.58/4.26 | 0.50/5.14 |
| 69 | xaemcq | 0.0311 | 348.4 | 24.02 | 874.0 | 0.50/5.14 | 0.37/2.85 | 0.58/4.26 | 0.50/5.14 |
| 70 | xaemcr | 0.0341 | 325.5 | 22.62 | 625.0 | 0.50/5.14 | 0.37/2.85 | 0.58/4.26 | 0.50/5.14 |
| 71 | xaemcs | 0.0133 | 305.8 | 23.98 | 904.8 | 0.50/5.14 | 0.37/2.85 | 0.58/4.26 | 0.44/4.58 |
| 72 | xaemct | 0.0300 | 22.0 | 22.82 | 959.2 | 0.50/5.14 | 0.37/2.85 | 0.58/4.26 | 0.44/4.58 |
| 73 | xaemcu | 0.0113 | 38.9 | 22.66 | 758.2 | 0.50/5.14 | 0.37/2.85 | 0.58/4.26 | 0.44/4.58 |
| 74 | xaemcv | 0.0495 | 81.6 | 24.44 | 830.4 | 0.50/5.14 | 0.37/2.85 | 0.58/4.26 | 0.44/4.58 |
| 75 | xaemcw | 0.0304 | 175.5 | 23.18 | 587.3 | 0.50/5.14 | 0.37/2.85 | 0.58/4.26 | 0.44/4.58 |
| 76 | xaemcx | 0.0411 | 216.0 | 23.08 | 575.0 | 0.24/3.05 | 0.20/2.01 | 0.31/2.53 | 0.33/3.71 |
| 77 | xaemcy | 0.0412 | 191.5 | 22.45 | 925.4 | 0.24/3.05 | 0.20/2.01 | 0.31/2.53 | 0.33/3.71 |
| 78 | xaemcz | 0.0374 | 215.0 | 24.19 | 597.9 | 0.24/3.05 | 0.20/2.01 | 0.31/2.53 | 0.33/3.71 |
| 79 | xaemda | 0.0150 | 270.7 | 22.36 | 856.4 | 0.24/3.05 | 0.20/2.01 | 0.31/2.53 | 0.33/3.71 |
| 80 | xaemdb | 0.0343 | 355.2 | 23.51 | 1054.3 | 0.24/3.05 | 0.20/2.01 | 0.31/2.53 | 0.33/3.71 |
| 81 | xaemdc | 0.0303 | 357.7 | 23.45 | 896.9 | 0.24/3.05 | 0.20/2.01 | 0.31/2.53 | 0.24/3.05 |
| 82 | xaemdd | 0.0091 | 311.1 | 23.16 | 716.8 | 0.24/3.05 | 0.20/2.01 | 0.31/2.53 | 0.24/3.05 |
| 83 | xaemde | 0.0109 | 80.9 | 23.73 | 993.9 | 0.24/3.05 | 0.20/2.01 | 0.31/2.53 | 0.24/3.05 |
| 84 | xaemdf | 0.0075 | 243.9 | 22.63 | 611.0 | 0.24/3.05 | 0.08/1.00 | 0.12/2.53 | 0.24/3.05 |
| 85 | xaemdg | 0.0609 | 267.5 | 24.23 | 844.0 | 0.24/3.05 | 0.08/1.00 | 0.12/1.45 | 0.24/3.05 |
| 86 | xaemdh | 0.0372 | 274.3 | 22.99 | 652.2 | 0.24/3.05 | 0.08/1.00 | 0.12/1.45 | 0.18/2.34 |
| 87 | xaemdi | 0.0214 | 314.1 | 22.86 | 1061.7 | 0.24/3.05 | 0.08/1.00 | 0.12/1.45 | 0.18/2.34 |
| 88 | xaemdj | 0.0432 | 76.8 | 24.14 | 868.4 | 0.24/3.05 | 0.08/1.00 | 0.12/1.45 | 0.18/2.34 |
| 89 | xaemdk | 0.0321 | 92.0 | 23.83 | 706.6 | 0.02/0.66 | 0.08/1.00 | 0.12/1.45 | 0.18/2.34 |
| 90 | xaemdl | 0.0389 | 109.5 | 23.03 | 691.8 | 0.02/0.66 | 0.08/1.00 | 0.12/1.45 | 0.18/2.34 |
| 91 | xaemdm | 0.0167 | 175.2 | 23.77 | 832.0 | 0.02/0.66 | 0.08/1.00 | 0.12/1.45 | 0.10/1.45 |
| 92 | xaemdn | 0.0406 | 103.9 | 22.42 | 791.0 | 0.02/0.66 | 0.08/1.00 | 0.12/1.45 | 0.10/1.45 |
| 93 | xaemdo | 0.0420 | 229.6 | 23.59 | 942.0 | 0.02/0.66 | 0.01/0.18 | 0.01/0.18 | 0.10/1.45 |
| 94 | xaemdp | 0.0335 | 250.9 | 23.46 | 988.7 | 0.02/0.66 | 0.01/0.18 | 0.01/0.18 | 0.10/1.45 |
| 95 | xaemdq | 0.0290 | 256.6 | 24.38 | 971.6 | 0.02/0.66 | 0.01/0.18 | 0.01/0.18 | 0.10/1.45 |
| 96 | xaemdr | 0.0080 | 76.4 | 24.37 | 765.1 | 0.02/0.66 | 0.01/0.18 | 0.01/0.18 | 0.02/0.66 |
| 97 | xaemds | 0.0183 | 53.5 | 22.60 | 607.5 | 0.02/0.66 | 0.01/0.18 | 0.01/0.18 | 0.02/0.66 |
| 98 | xaemdt | 0.0300 | 83.4 | 22.11 | 929.8 | 0.02/0.66 | 0.01/0.18 | 0.01/0.18 | 0.02/0.66 |
| 99 | xaemdu | 0.0416 | 96.3 | 23.80 | 1021.3 | 0.02/0.66 | 0.01/0.18 | 0.01/0.18 | 0.02/0.66 |
| 100 | xaemdv | 0.0073 | 126.9 | 23.41 | 562.2 | 0.02/0.66 | 0.01/0.18 | 0.01/0.18 | 0.02/0.66 |

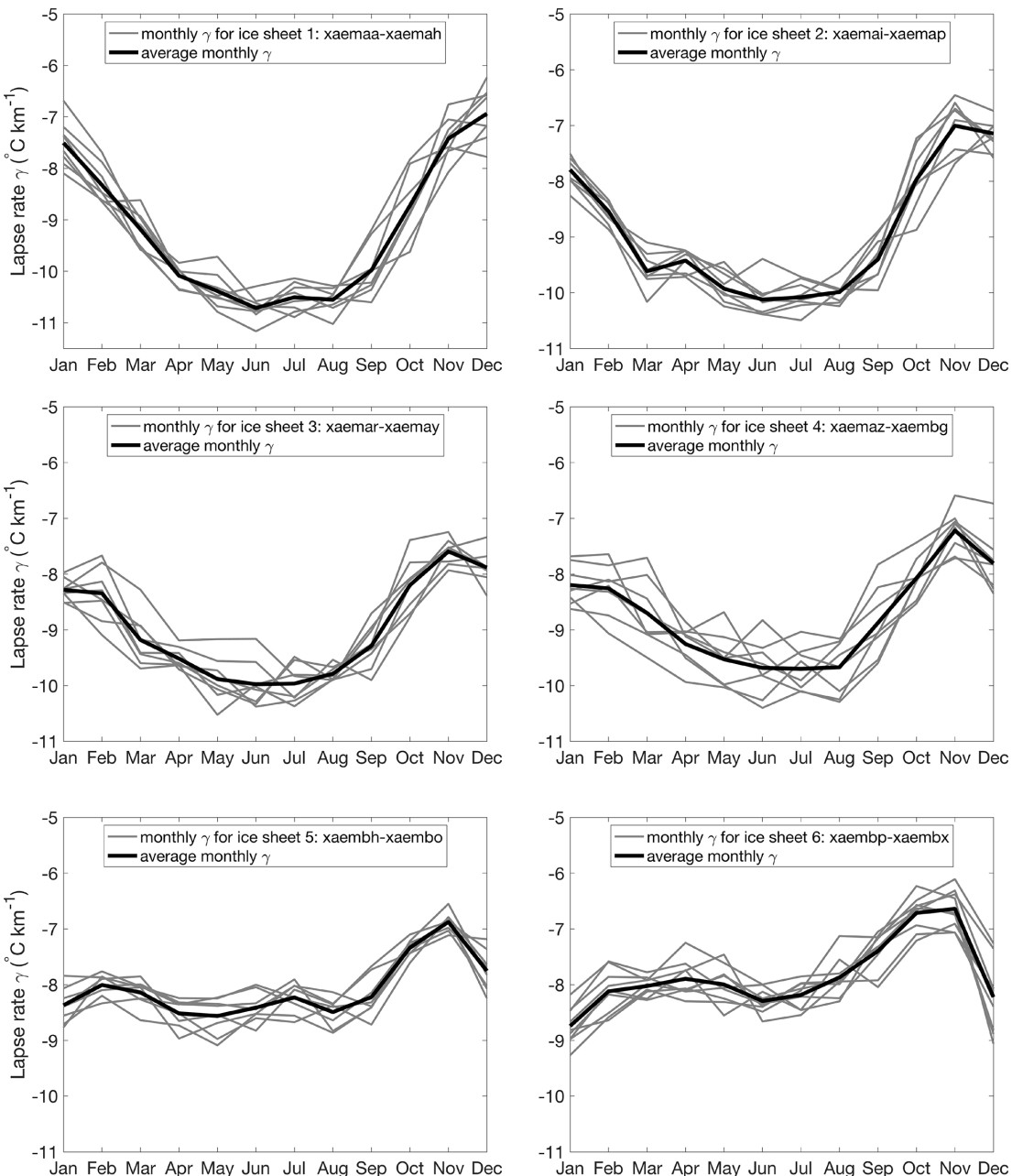

**Figure A1.**

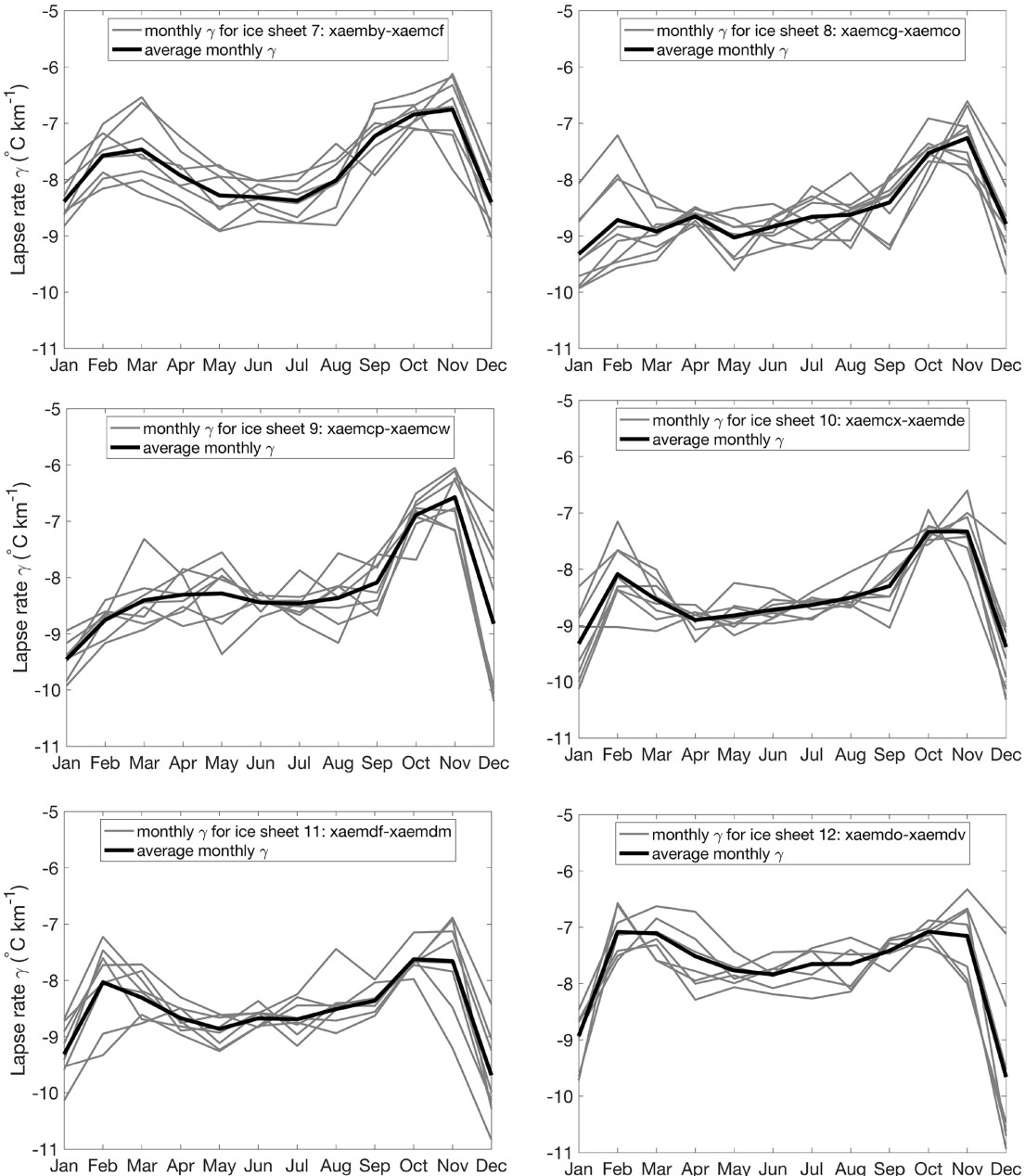

**Figure A1.** Monthly average lapse rate for the 12 different ice sheet geometries calculated from the 100 GCM runs for EMULATOR_12b.

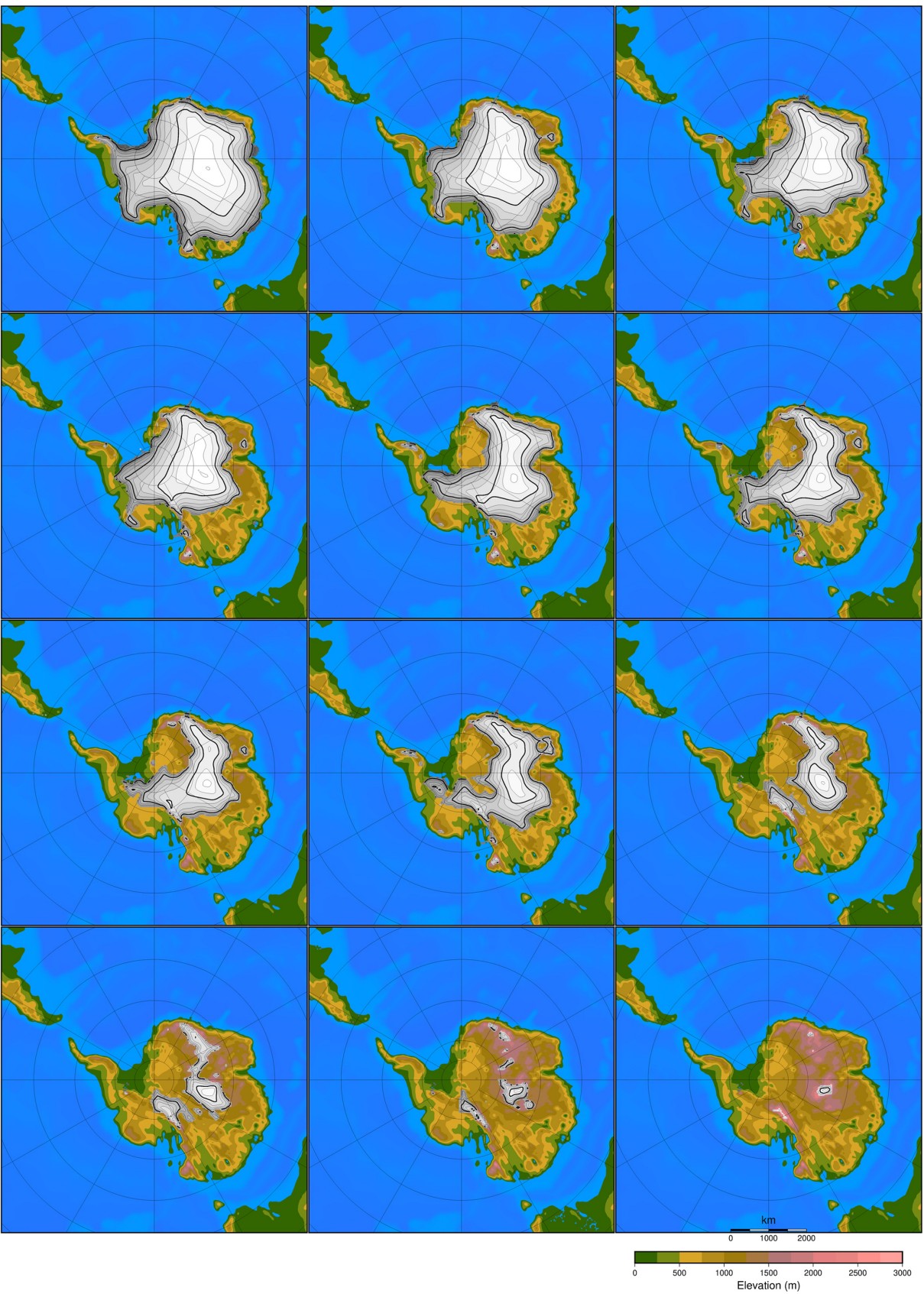

**Figure A2.** The 12 ice sheet geometries used as input to EMULATOR_12a.

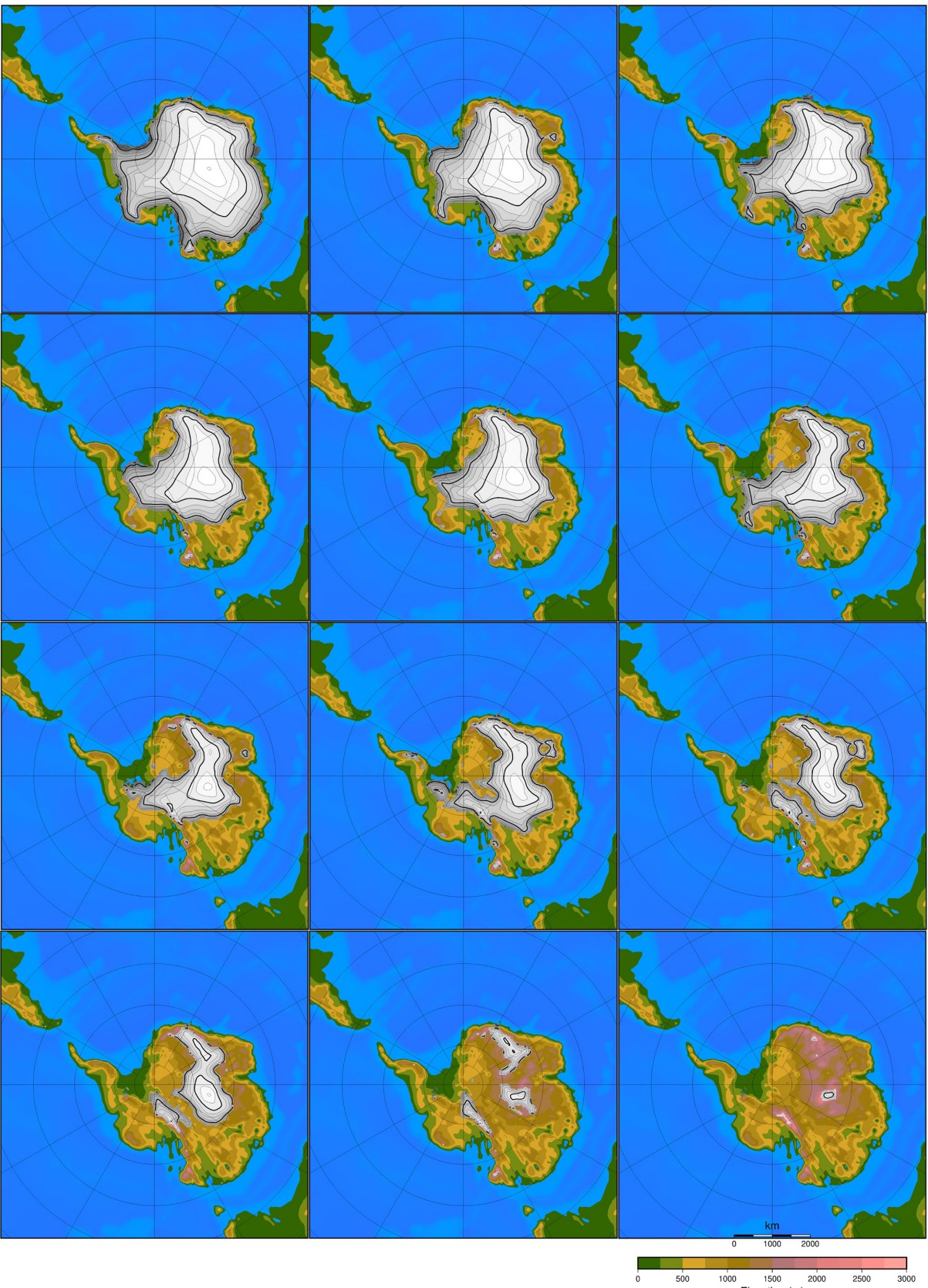

**Figure A3.** The 12 ice sheet geometries used as input to EMULATOR_12b.

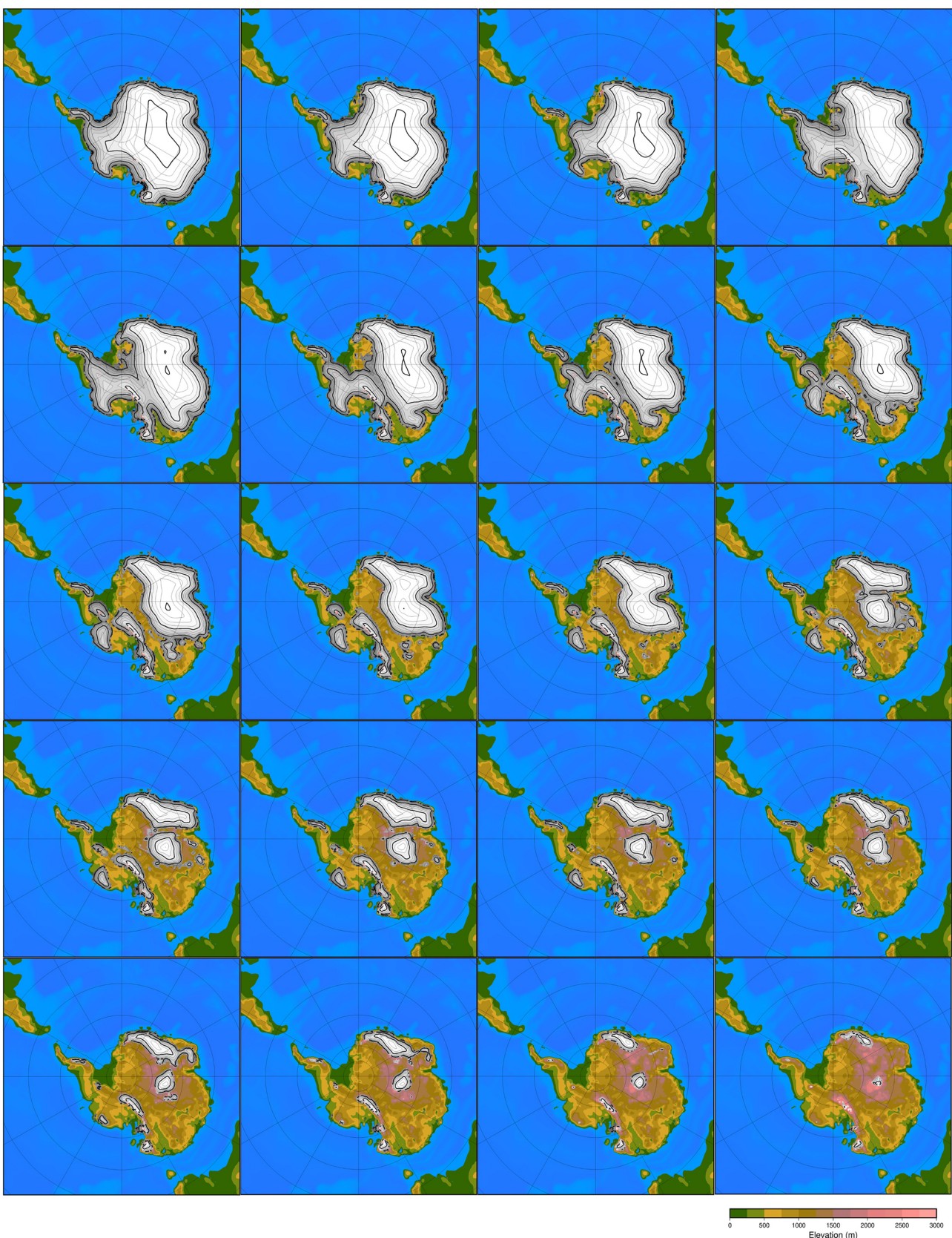

**Figure A4.** The 20 ice sheet geometries used as input to EMULATOR_20.

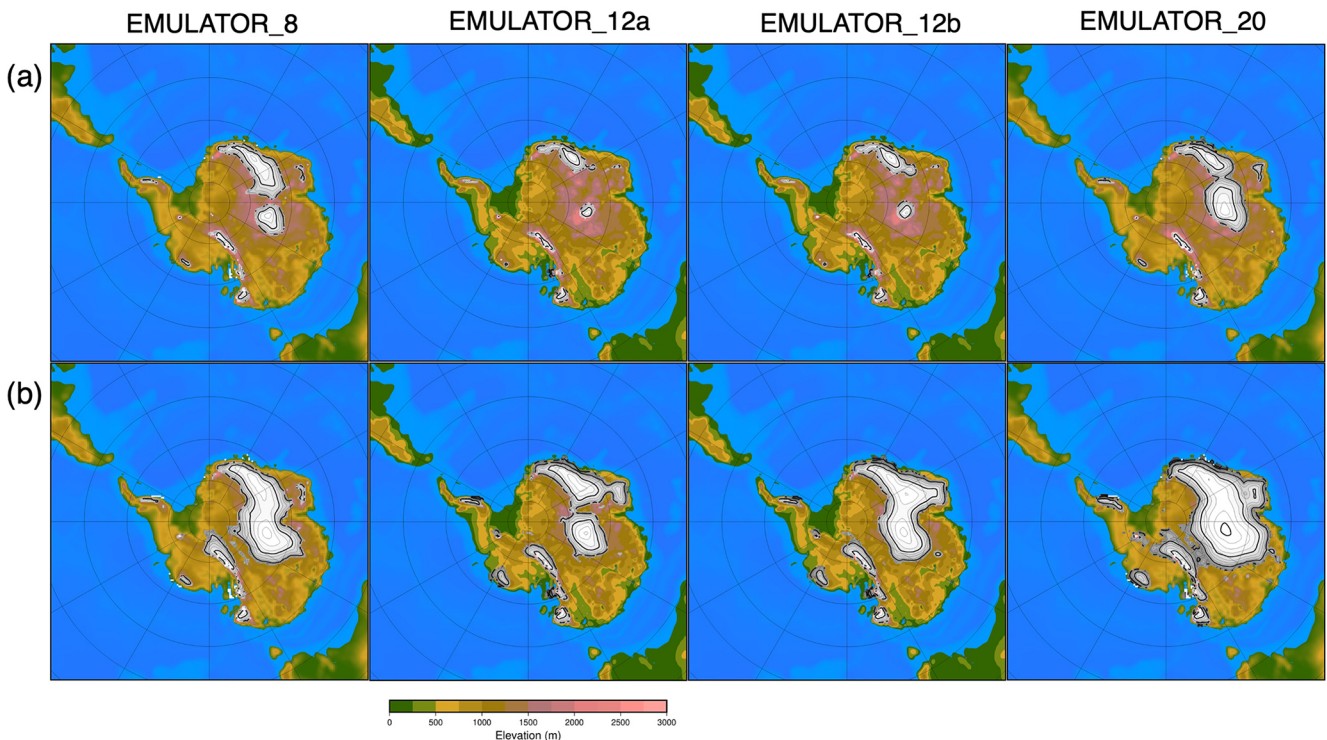

**Figure A5.** Snapshots of the ice sheet geometry after **(a)** 47 kyr and **(b)** 60 kyr (corresponding to the temperature fields in Fig. 10) for EMULATOR_8, EMULATOR_12a, EMULATOR_12b and EMULATOR_20 calibrated with ice volume.

*Code and data availability.* The code for the coupler CLISEMv1.0 between the climate and the ice sheet model is available on Zenodo: https://doi.org/10.5281/zenodo.5245156 (Van Breedam et al., 2021a). All data used in this paper are available upon request.

*Video supplement.* A video supplement showing the ice sheet evolution for the three best-performing emulators calibrated on ice volume is available on Zenodo: https://doi.org/10.5281/zenodo.5242914 (Van Breedam et al., 2021b).

*Author contributions.* JVB designed the coupling method between the emulator and the ice sheet model. MC developed the Gaussian process emulator. PH developed the ice sheet model code. JVB wrote the manuscript with contributions from all co-authors.

*Competing interests.* Philippe Huybrechts is topical editor of *Geoscientific Model Development*.

*Acknowledgements.* We would like to thank two anonymous reviewers for their detailed comments and useful suggestions to improve the manuscript. Michel Crucifix is funded as Research Director with the Belgian National Fund of Scientific Research FNRS.

*Financial support.* This research has been supported by the NAME OF FUNDER (grant no. GRANT AGREEMENT NO). TS6

*Review statement.* This paper was edited by Alexander Robel and reviewed by two anonymous referees.

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

**Remarks from the typesetter**

TS1    Several instances (like $f(\boldsymbol{x}_i)$ in this case or most instances in Eqs. 6–8) were already formatted as vectors. Since you only highlighted them but did not provided any explanations, it was not clear if anything in $f(\boldsymbol{x}_i)$ needed to be changed. Furthermore, you did not indicate in your highlights whether a vector or a matrix was meant. I still tried to identify them, please carefully check.

TS2    Please clarify if I and V are vectors or matrices.

TS3    Yes, all changes in equations require the editor's approval. Therefore, please provide a detailed explanation for those changes, so we can start the post-review adjustments process. If you still wish to insert the definition for I, please add it to your request for the editor.

TS4    Please clarify if A, H, V and P are vectors or matrices.

TS5    Please clarify if T is a vector or a matrix.

TS6    Please note that the financial support section is mandatory if funding information is provided. However, it is possible – but not necessary – to remove repeated information from the acknowledgements.