# Peer review of "A Gaussian process emulator for simulating ice sheet-climate interactions on a multi-million year timescale: CLISEMv1.0"

_Geoscientific Model Development, 2021_

## Referee Comment (RC1)

**Review Report on "A Gaussian process emulator for simulating ice sheet-climate interactions on a multi-million year timescale: CLISEMv1.0"**

The manuscript describes a coupled emulation approach called CLISEMv1.0, which builds two separate emulators for an ice sheet model (AISMPALEO) and a climate model (HadSM3) – The outputs from these two emulators provide inputs to each other, enabling synchronous simulation of ice sheet evolution and climate changes. The authors conducted several sensitivity analyses regarding how the two emulators are built (e.g. depending on how the ice sheet input for the climate emulator is defined), how long the coupling time is, and how the lapse rate is adjusted to account for the elevation difference between the climate model grid and the ice model grid. While the coupled emulation approach itself is scientifically highly important and perhaps long overdue, the current coupled emulation results shown in Sections 3 and 4 need a lot of further clarification before the manuscript can be considered for being published in GMD. More specific comments are listed below.

**Major comments**:

1. The description about the three different experimental design setups (EMULATOR_70, EMULATOR_100a, and EMULATOR_100b) in Section 2.3 and the result of coupled experiments described in Section 3.1 are contradicting to each other. According to Section 2.3, "EMULATOR_70 has a good spread between the different ice volumes and ice areas" but somehow the results described in Section 3.1 and also shown in Figure 9 indicate that the design EMULATOR_70 seems to have some serious flaw. The other two are described as designs with some notable flaws in Section 2.3, but somehow lead to better results. The manuscript gives some brief description on this issue in Section 3.1, but the authors did not really get to the bottom of the issue – In fact I cannot find any good rationale for how the ice model settings for EMULATOR_100a, and EMULATOR_100b are determined at the beginning – Why did the author decided to let EMULATOR_100a have "more small ice sheet geometries (ice volumes) compared to EMULATOR_100b (Figure 4) and has a good spread for the ice area of the input ice sheet geometries" and EMULATOR_100b be "poorly defined by ice sheet area as there are several experiments with the same ice sheet area yet different ice sheet geometry, but is well defined for ice sheet volume"? Are they some data of opportunity from some other experiments? Or did the author gradually add more model runs to these to designs until they give some sensible results shown in Figure 9? I think the authors need to describe their decision making process behind these design points in detail.

2. Related to the above point, it is hard for me to understand why the coupled emulation based on EMULATOR_70 leads to such poor results. For example, why does it lead to ice volume change that is largely unresponsive to the $CO_2$ concentration change when ice volume is used? Similarly, to me it is hard to figure out the true reason for the poor results in Figure 9c for all three design schemes. If an emulator based on two parameters (ice volume and area in this case) leads to a worse result than an emulator based on only one parameter for the *same perturbed physics ensemble*, the only possible explanation is that the emulator with the two parameters failed to capture the behavior of the original simulator. I suspect the poor results stem from the poor emulation accuracy when the emulators are built on both ice volume and ice area. In fact, Table 1a shows that there are only **11** design points for the two ice parameters (ice area and ice volume) and, to make the problem worse, these two parameters are highly correlated. I think any emulation approaches are destined to fail with such a small number of design points that are highly correlated with each other.

3. I am not sure why Section 4 is called 'Bayesian' sensitivity analysis because nothing in the section seems to be particularly 'Bayesian'. There seems to be no consideration on uncertainty in the model parameters in the form of posterior densities, which is typically done in Bayesian analysis. In addition, the authors somehow decided to throw away the emulators and build a new time series model for uncertainty quantification. Is there any particular reason behind this decision? I am also wondering if there is any particular reason that only the first-order autocorrelation is considered here.

**Minor Comments**:

1. Lines 204-215: Related to the Major Comment 1 above, I think describing EMULATOR_100a and EMUATOR_100b as 'bad' emulator seems to be weird. I think this part can be improved by clarifying how EMULATOR_100a and EMULATOR_100b are actually designed; otherwise, readers may wonder why the authors decided to use 'bad designs'. Later, in Section 3, they will be surprised by the fact that these 'bad designs' led to better results.

2. Lines 239-240 "Therefore, they might be doing a poor job in reconstructing the simulated temperatures well.": I think 'well' should be deleted.

3. Line 259-260: "The notion of ice sheet parameter as an emulator input is introduced in previous studies to be an integer ranging from 1 to the number of ice sheet geometries": The sentence does not make much sense. Please revise.

4. I feel that the overall writing quality of Sections 3 and 4 are notably worse than that of the other Sections. Hopefully the authors can improve the texts in the revised version.

---

## Author Comment (AC1)

**Response to Anonymous Referee #1**

The manuscript describes a coupled emulation approach called CLISEMv1.0, which builds two separate emulators for an ice sheet model (AISMPALEO) and a climate model (HadSM3) – The outputs from these two emulators provide inputs to each other, enabling synchronous simulation of ice sheet evolution and climate changes. The authors conducted several sensitivity analyses regarding how the two emulators are built (e.g. depending on how the ice sheet input for the climate emulator is defined), how long the coupling time is, and how the lapse rate is adjusted to account for the elevation difference between the climate model grid and the ice model grid. While the coupled emulation approach itself is scientifically highly important and perhaps long overdue, the current coupled emulation results shown in Sections 3 and 4 need a lot of further clarification before the manuscript can be considered for being published in GMD. More specific comments are listed below.

**Author's response:** We thank the reviewer for his critical analysis, which has definitely helped us to improve the clarity and quality of the manuscript. In the revised version, we give more explanations about why certain model set-up choices are made. Also, we have been trying to be more precise about the influence of this choices on the model performance.

**Major comments**:
1.  The description about the three different experimental design setups (EMULATOR_70, EMULATOR_100a, and EMULATOR_100b) in Section 2.3 and the result of coupled experiments described in Section 3.1 are contradicting to each other. According to Section 2.3, "EMULATOR_70 has a good spread between the different ice volumes and ice areas" but somehow the results described in Section 3.1 and also shown in Figure 9 indicate that the design EMULATOR_70 seems to have some serious flaw. The other two are described as designs with some notable flaws in Section 2.3, but somehow lead to better results. The manuscript gives some brief description on this issue in Section 3.1, but the authors did not really get to the bottom of the issue – In fact I cannot find any good rationale for how the ice model settings for EMULATOR_100a, and EMULATOR_100b are determined at the beginning – Why did the author decided to let EMULATOR_100a have "more small ice sheet geometries (ice volumes) compared to EMULATOR_100b (Figure 4) and has a good spread for the ice area of the input ice sheet geometries" and EMULATOR_100b be "poorly defined by ice sheet area as there are several experiments with the same ice sheet area yet different ice sheet geometry, but is well defined for ice sheet volume"? Are they some data of opportunity from some other experiments? Or did the author gradually add more model runs to these to designs until they give some sensible results shown in Figure 9? I think the authors need to describe their decision making process behind these design points in detail.

**Author's response:** We agree that the explanation for the 'bad' and 'good' performance of the emulator was not stated in a way that was clear enough for the reader. The performance of the emulator is primarily dependent on the number of ice sheet geometries the emulator is calibrated on and on the spacing of the ice sheet geometries. Other tuning factors are of secondary importance. Even though EMULATOR_70 has a good spread for the different ice sheet geometries concerning ice volume and ice area, the eight different input ice sheet geometries are not enough to initiate the transition to a continental scale glaciation. EMULATOR_100a and EMULATOR_100b - both calibrated on precursor climate model runs including twelve different ice sheet geometries appeared to be sufficient to induce the transition to glacial conditions.

To increase the clarity of the manuscript with respect to the different model choices made and the model performance, we introduce in the revised manuscript another emulator in which 100 climate model runs are run based on 20 different input ice sheet geometries with a good spread for both ice volume and ice area. This way, it is tested whether adding more ice sheet geometries is improving the model performance. The number of different input ice sheet geometries is limited by the climate model resolution, adding more geometries only adds information if a grid point of the climate model changes from tundra to ice. Using 20 different input ice sheet geometries, we capture the latitudinal grid spacing of 2.5° or 277.5 km for a span of maximum 4500 km (the longitudinal grid spacing at 60°S - the edge of the Antarctic continent in our simulations - is 55.6 km and decreases towards the pole). The names of

the different emulators have also been changed to EMULATOR_8, EMULATOR_12a, EMULATOR_12b and EMULATOR_20, indicating the number of ice sheet geometries used for the precursor climate model runs instead of the number of climate model runs performed (which is less relevant). The reason to include another emulator is that the number of ice sheet geometries has a quite large influence on whether the glacial inception will occur or not. While using a different number of ice sheet geometries, it is tested how many of these prescribed ice sheet geometries are needed to make the emulator work properly. Since the number of climate model runs is not crucial in the performance of the model, EMULATOR_70 with 8 input ice sheet geometries  and based on 70 climate model runs is replaced by EMULATOR_8, also with 8 input ice sheet geometries and based on 100 different climate model runs.

2. Related to the above point, it is hard for me to understand why the coupled emulation based on EMULATOR_70 leads to such poor results. For example, why does it lead to ice volume change that is largely unresponsive to the CO2 concentration change when ice volume is used? Similarly, to me it is hard to figure out the true reason for the poor results in Figure 9c for all three design schemes. If an emulator based on two parameters (ice volume and area in this case) leads to a worse result than an emulator based on only one parameter for the *same perturbed physics ensemble*, the only possible explanation is that the emulator with the two parameters failed to capture the behavior of the original simulator. I suspect the poor results stem from the poor emulation accuracy when the emulators are built on both ice volume and ice area. In fact, Table 1a shows that there are only **11** design points for the two ice parameters (ice area and ice volume) and, to make the problem worse, these two parameters are highly correlated. I think any emulation approaches are destined to fail with such a small number of design points that are highly correlated with each other.

**Author's response**: The answer is partially given in the previous response, but extended here. EMULATOR_70, calibrated on eight different initial ice sheet geometries appeared to be insufficient to induce the glacial transition. The climate simulated with a certain ice sheet geometry has a very sharp boundary at the edge of the ice sheet because of the large difference in albedo between ice and tundra. If you have insufficient ice sheet geometries the emulator will not simulate a climate that is cold enough in order to reach the climatic imprint as simulated by the next, larger ice sheet geometry.

'*Figure 9a shows the ice sheet evolution for the four emulators calibrated on ice volume. EMULATOR_12a, EMULATOR_12b and EMULATOR_20 show the transition towards a continental scale ice sheet in a very narrow $CO_2$ interval of 845 to 875 ppmv. On the other hand, EMULATOR_8 does not seem to show any sensitivity to the $CO_2$ forcing during the 3 million year-long simulation (and also not on a longer timescale). The reason is that the prescribed ice sheets are separated too much in the initial climate model runs. Because of the large difference in albedo between ice and tundra, the prescribed ice sheets create a sharp boundary at the ice sheet margin that is visible in the temperature field. If insufficient prescribed ice sheet geometries are used, the ice sheet does not grow enough to reach the next input ice sheet geometry and the emulated temperatures remain too warm at the ice sheet margin. It appears that the threshold on the number of needed ice sheet geometries is somewhere between 8 and 12. Using 20 input ice sheet geometries does not lead to a significant improvement in the model performance.*'

'*Another option is to calibrate the emulator based on the ice area, which has a direct influence on the albedo. The glaciation threshold for EMULATOR_12a and EMULATOR_20 shows a very similar sensitivity to $CO_2$ of about 860 ppmv (Figure 9b). EMULATOR_8 grows immediately to a medium sized ice sheet and cannot grow further towards a continental scale ice sheet for the reasons already quoted. EMULATOR_12b was poorly defined on ice area (several ice sheet geometries had a similar area, but different geometry) and the ice sheet grows immediately towards a continental scale. Therefore, in addition to having sufficient ice sheet geometries, a good spacing of the ice sheet parameter is another requirement to use an emulator for coupled ice sheet-climate simulations.*'

Table A1 shows **100** design points (each design point has a value for the orbital parameters, $CO_2$ and the ice sheet parameter) and contains **12** values for the ice sheet parameter. Indeed there is a strong correlation between the ice sheet parameters ice volume and ice area, but not for every ice sheet geometry as ice sheets can have similar volumes for different areas or vice versa. The reason why the emulator based on ice area and ice volume is performing worse compared to the other emulators, is because the tuned correlation length scales are much lower for the ice sheet parameters and therefore less decisive in the climate signal compared to the $CO_2$ forcing and the orbital forcing (because of the contradictory values with similar ice sheet volume, yet different ice sheet area).

It is true that the poor results stem from the poor emulation accuracy. But the poor emulation accuracy originates from the ambiguous signal of the ice sheet parameter: when ice volume increases with a certain magnitude, the ice area will respond non-linear. When the emulator is tuned based on both ice area and ice volume, there are 6 variables while only 5 parameters are actually different in the forcing of the climate. We think it is an interesting point to raise that more information on the ice sheets is not leading to a better performance of the emulator.

*'When the coupling is based on both the ice volume and the ice area, the simulated ice sheet volume mostly follows the insolation pattern (Figure 9c). Remarkably, the overall simulated ice volumes stay small. The emulator calibrated on both ice area and ice volume has a low correlation length scale for the ice sheet parameter in comparison to the orbital parameters and $CO_2$, suggesting that there is a weak relationship between temperature changes and ice sheet size. The poor emulation accuracy originates from calibrating the emulator based on 6 variables, while actually only 5 input forcing parameters are variable. The additional information on the ice sheet parameter is strongly correlated in most cases, but arises to give an ambiguous climate signal because the spread between ice volume and ice area is not equal. Thus, calibrating the emulator with additional information on the ice sheet parameter does not lead to a better performance of the coupled ice sheet-climate simulations.'*

Additionally, we devoted a paragraph in the discussion about the problems that a double ice sheet parameter induces.

*'A common problem for the emulator and the matrix look-up table method, where the ice sheet parameter is defined by a single number, is that there is no control on the regions where ice starts to grow. There are no obvious solutions to this problem. Adding an additional regional ice sheet parameter, that would receive a certain value depending on whether the region is ice-covered or not, would give rise to an ice sheet parameter that is defined in multiple ways. It has been shown that calibrating the emulator on two parameters (ice area and ice volume) that are strongly correlated and not independent, induces problems with the emulation accuracy. The best solution to overcome the lack of spatial control on ice sheet growth is to use enough prescribed ice sheet geometries in the model design, in order to give limited degrees of freedom to the emulator.'*

3. I am not sure why Section 4 is called 'Bayesian' sensitivity analysis because nothing in the section seems to be particularly 'Bayesian'. There seems to be no consideration on uncertainty in the model parameters in the form of posterior densities, which is typically done in Bayesian analysis. In addition, the authors somehow decided to throw away the emulators and build a new time series model for uncertainty quantification. Is there any particular reason behind this decision? I am also wondering if there is any particular reason that only the first-order autocorrelation is considered here.

**Author's response:** In section 4, the sensitivity of the ice sheet evolution is tested based on Bayesian statistics, i.e. the climate is predicted, taken into account the climate at the previous time step. In that sense, we believe that the use of Bayesian sensitivity analysis is legitimate. We do not investigate the influence of the climate evolution based on different model parameters, but the sensitivity to having prior knowledge about the climate that was simulated one coupling time step before. The reason to only include the first order auto-correlation is for simplicity and justified because the first order auto-correlation captures the behaviour well (when the climate was cold in the previous time-step, the chance is high that the climate was also cold in the time step before because the climatic changes occur gradually).

The emulator is akin to the emulators shown in section 3, with the emulation of precipitation and January temperature exactly the same way as in the other experiments. Since only the uncertainty of one of the emulated fields can be investigated at once, the parameter most decisive for ice sheet growth is chosen. It appears to be temperature, especially summer temperature and hence uncertainty to January temperature is emulated using a Bayesian approach.

*'The additional value of the use of an emulator for coupled ice sheet-climate simulations is that the mean climate predictions come with the estimate of its variance. In that sense, the Bayesian framework of the emulator can be used to update the covariance between two climate prediction points to predict the variance of the next point, knowing the variance of the current state. In this section, the Bayesian framework of the emulator is used to explore the uncertainty of the emulator by performing 50 Monte Carlo simulations including the variance on EMULATOR_12b calibrated with ice volume.'*

*'The emulator is exactly similar to EMULATOR_12b, except for the inclusion of the variance for the temperature fields. The uncertainty is explored, taking into account the correlation between the temperature field at the previous time step and the current emulated temperature field. The Gaussian process emulator has an exponential decaying correlation function. Most of the covariance structure is captured by the first-order autocorrelation. The climatic changes occur gradually, a cold climate state will not be preceded by a very warm climate and therefore, the first-order autocorrelation captures the behaviour well.'*

**Minor Comments**:
1. Lines 204-215: Related to the Major Comment 1 above, I think describing EMULATOR_100a and EMUATOR_100b as 'bad' emulator seems to be weird. I think this part can be improved by clarifying how EMULATOR_100a and EMUATOR_100b are actually designed; otherwise, readers may wonder why the authors decided to use 'bad designs'. Later, in Section 3, they will be surprised by the fact that these 'bad designs' led to better results.

**Author's response:** We tried to clarify what is investigated with the different emulators and what is a 'good' design and a 'bad' design. Clearly, bad designs do not lead to good results and oppositely, good designs (in terms of ice sheet spacing) can only lead to bad results when insufficient ice sheet geometries are included.

Lines 239-240 "Therefore, they might be doing a poor job in reconstructing the simulated temperatures well.": I think 'well' should be deleted.

**Author's response:** Done.

2. Line 259-260: "The notion of ice sheet parameter as an emulator input is introduced in previous studies to be an integer ranging from 1 to the number of ice sheet geometries": The sentence does not make much sense. Please revise.

**Author's response:** The sentence is revised to: *'The ice sheet parameter is a single number representing the shape and area of the prescribed ice sheets. In previous studies (Araya-Melo et al., 2015; Lord et al., 2017), it is defined as an integer, representing the number of different ice sheet geometries.'*

3. I feel that the overall writing quality of Sections 3 and 4 are notably worse than that of the other Sections. Hopefully the authors can improve the texts in the revised version.

**Author's response:** We acknowledge that most comments were related to the model set-up and the model performance. The text has been revised by stating why certain decisions have been made or what certain decisions have learned us. We have tried to improve the clarity of the manuscript by including an additional emulator with 20 prescribed ice sheet geometries, since the number of prescribed ice sheet geometries has a main influence on the emulator performance. We also rewrote the critical paragraphs from section 3, including the results with the additional EMULATOR_20 and gave arguments why the use of the Bayesian sensitivity analysis in section 4 was justified.

---

## Author Response (AR1)

**Response to Anonymous Referee #1**

The manuscript describes a coupled emulation approach called CLISEMv1.0, which builds two separate emulators for an ice sheet model (AISMPALEO) and a climate model (HadSM3) – The outputs from these two emulators provide inputs to each other, enabling synchronous simulation of ice sheet evolution and climate changes. The authors conducted several sensitivity analyses regarding how the two emulators are built (e.g. depending on how the ice sheet input for the climate emulator is defined), how long the coupling time is, and how the lapse rate is adjusted to account for the elevation difference between the climate model grid and the ice model grid. While the coupled emulation approach itself is scientifically highly important and perhaps long overdue, the current coupled emulation results shown in Sections 3 and 4 need a lot of further clarification before the manuscript can be considered for being published in GMD. More specific comments are listed below.

**Author's response:** We thank the reviewer for his critical analysis, which has definitely helped us to improve the clarity and quality of the manuscript. In the revised version, we give more explanations about why certain model set-up choices are made. Also, we have been trying to be more precise about the influence of this choices on the model performance.

**Major comments**:
1. The description about the three different experimental design setups (EMULATOR_70, EMULATOR_100a, and EMULATOR_100b) in Section 2.3 and the result of coupled experiments described in Section 3.1 are contradicting to each other. According to Section 2.3, "EMULATOR_70 has a good spread between the different ice volumes and ice areas" but somehow the results described in Section 3.1 and also shown in Figure 9 indicate that the design EMULATOR_70 seems to have some serious flaw. The other two are described as designs with some notable flaws in Section 2.3, but somehow lead to better results. The manuscript gives some brief description on this issue in Section 3.1, but the authors did not really get to the bottom of the issue – In fact I cannot find any good rationale for how the ice model settings for EMULATOR_100a, and EMULATOR_100b are determined at the beginning – Why did the author decided to let EMULATOR_100a have "more small ice sheet geometries (ice volumes) compared to EMULATOR_100b (Figure 4) and has a good spread for the ice area of the input ice sheet geometries" and EMULATOR_100b be "poorly defined by ice sheet area as there are several experiments with the same ice sheet area yet different ice sheet geometry, but is well defined for ice sheet volume"? Are they some data of opportunity from some other experiments? Or did the author gradually add more model runs to these to designs until they give some sensible results shown in Figure 9? I think the authors need to describe their decision making process behind these design points in detail.

**Author's response:** We agree that the explanation for the 'bad' and 'good' performance of the emulator was not stated in a way that was clear enough for the reader. The performance of the emulator is expectedly dependent on the experiment design, and in fact, primarily dependent on the number of ice sheet geometries the emulator is calibrated on and on the spacing of the ice sheet geometries. Other factors are of secondary importance. Even though EMULATOR_70 has a good spread for the different ice sheet geometries concerning ice volume and ice area, the eight different input ice sheet geometries are not sufficient to initiate the transition to a continental scale glaciation. EMULATOR_100a and EMULATOR_100b - both calibrated on precursor climate model runs including twelve different ice sheet geometries appeared to be sufficient to induce the transition to glacial conditions.

To increase the clarity of the manuscript with respect to the different model choices made and the model performance, we introduce in the revised manuscript another emulator in which 100 climate model runs are run based on 20 different input ice sheet geometries with a good spread for both ice volume and ice area. This way, it is tested whether adding more ice sheet geometries is improving the model performance. The number of different input ice sheet geometries is limited by the climate model resolution, adding more geometries only adds information if a grid point of the climate model changes from tundra to ice. Using 20 different input ice sheet geometries, we capture the latitudinal grid spacing of 2.5˚ or 277.5 km for a span of maximum 4500 km (the longitudinal grid spacing at 60˚S - the edge of the Antarctic continent in our simulations - is 55.6 km and decreases towards the pole). The names of

the different emulators have also been changed to EMULATOR_8, EMULATOR_12a, EMULATOR_12b and EMULATOR_20, indicating the number of ice sheet geometries used for the precursor climate model runs instead of the number of climate model runs performed (which is less relevant). The reason to include another emulator is that the number of ice sheet geometries has a quite large influence on whether the glacial inception will occur or not. While using a different number of ice sheet geometries, it is tested how many of these prescribed ice sheet geometries are needed to make the emulator work properly. Since the number of climate model runs is not crucial in the performance of the model, EMULATOR_70 with 8 input ice sheet geometries and based on 70 climate model runs is replaced by EMULATOR_8, also with 8 input ice sheet geometries and based on 100 different climate model runs.

*'Since the atmosphere-slab ocean model is time-efficient, it is chosen to run 100 climate model runs with 5 variable forcing parameters. The ice sheets have a very distinct climatic imprint compared to the orbital parameters and the $CO_2$ level, which all result in smooth climatic fields. Because of the large difference in albedo between ice and tundra at the edge of the ice sheet, the climatic imprint of a certain ice sheet geometry has a sharp boundary. The number of ice sheet geometries taken into the model design of the emulator might therefore have a large impact on the performance of the emulator. To test the impact of the ice sheet parameter, four different emulators are constructed based on a different number of predefined ice sheet geometries or based on a different spread of the ice sheet geometries. The different emulators are named according to the number of ice sheet geometries in the model design, being 8, 12 and 20 for respectively EMULATOR_8, EMULATOR_12a, EMULATOR_12b and EMULATOR_20.*

*Except for the number of predefined ice sheet geometries, also the spread of the different prescribed geometries is varying between the different emulators, depending on whether the ice sheet parameter is defined by ice area or by ice volume. EMULATOR_8, EMULATOR_12a and EMULATOR_20 have a good spread between the different ice sheet geometries in terms of ice volume and ice area. EMULATOR_12b is well defined for ice volume, but poorly defined by ice sheet area as there are several experiments with the same ice sheet area yet different ice sheet geometry. (see Table A1 in Appendix for the experimental parameter values). This way, the influence of the spread in ice sheet volume/ice area on the emulated climate is investigated. The spacing of the different ice sheet geometries is expected to be crucial for medium sized ice sheets, because they constitute a transition zone towards a fully glaciated continent. EMULATOR_20 has the smallest spacing of ice sheet geometries around the crucial medium sized ice sheets, separated at the minimum distance that corresponds to the resolution of the climate model. The maximum ice sheet geometry in the model design for EMULATOR_12 is smaller than the maximum ice sheet geometry in the model design for EMULATOR_20 and EMULATOR_8. The objective designing EMULATOR_12 is to evaluate to what extent the emulator can still be used in an extrapolation regime beyond the largest ice sheet geometry.'*

2. Related to the above point, it is hard for me to understand why the coupled emulation based on EMULATOR_70 leads to such poor results. For example, why does it lead to ice volume change that is largely unresponsive to the CO2 concentration change when ice volume is used? Similarly, to me it is hard to figure out the true reason for the poor results in Figure 9c for all three design schemes. If an emulator based on two parameters (ice volume and area in this case) leads to a worse result than an emulator based on only one parameter for the *same perturbed physics ensemble*, the only possible explanation is that the emulator with the two parameters failed to capture the behavior of the original simulator. I suspect the poor results stem from the poor emulation accuracy when the emulators are built on both ice volume and ice area. In fact, Table 1a shows that there are only **11** design points for the two ice parameters (ice area and ice volume) and, to make the problem worse, these two parameters are highly correlated. I think any emulation approaches are destined to fail with such a small number of design points that are highly correlated with each other.

**Author's response**: The answer is partially given in the previous response, but extended here.

EMULATOR_70, calibrated on eight different initial ice sheet geometries appeared to be insufficient to induce the glacial transition. The simulated temperature field presents a very sharp boundary at the edge of the ice sheet because of the large difference in albedo between ice and tundra. If the training set is not dense enough the emulator will not simulate a climate that is cold enough in order to reach the climatic imprint as simulated by the next, larger ice sheet geometry.

'*Figure 9a shows the ice sheet evolution for the four emulators calibrated on ice volume. EMULATOR_12a, EMULATOR_12b and EMULATOR_20 show the transition towards a continental scale ice sheet in a very narrow $CO_2$ interval of 845 to 875 ppmv. On the other hand, EMULATOR_8 does not seem to show any sensitivity to the $CO_2$ forcing during the 3 million year-long simulation (and also not on a longer timescale). The reason is that the prescribed ice sheets are separated too much in the initial climate model runs. Because of the large difference in albedo between ice and tundra, the prescribed ice sheets create a sharp boundary at the ice sheet margin that is visible in the temperature field. If insufficient prescribed ice sheet geometries are used, the ice sheet does not grow enough to reach the next input ice sheet geometry and the emulated temperatures remain too warm at the ice sheet margin. It appears that the threshold on the number of needed ice sheet geometries is somewhere between 8 and 12. Using 20 input ice sheet geometries does not lead to a significant improvement in the model performance.*'

'*Another option is to calibrate the emulator based on the ice area, which has a direct influence on the albedo. The glaciation threshold for EMULATOR_12a and EMULATOR_20 shows a very similar sensitivity to $CO_2$ of about 860 ppmv (Figure 9b). EMULATOR_8 grows immediately to a medium sized ice sheet and cannot grow further towards a continental scale ice sheet for the reasons already quoted. EMULATOR_12b was poorly defined on ice area (several ice sheet geometries had a similar area, but different geometry) and the ice sheet grows immediately towards a continental scale. Therefore, in addition to having sufficient ice sheet geometries, a good spacing of the ice sheet parameter is another requirement to use an emulator for coupled ice sheet-climate simulations.*'

Table A1 shows **100** design points (each design point has a value for the orbital parameters, $CO_2$ and the ice sheet parameter) and contains **12** values for the ice sheet parameter. Indeed there is a strong correlation between the ice sheet parameters ice volume and ice area, but not for every ice sheet geometry as ice sheets can have similar volumes for different areas or vice versa.

Now we come to the point raised by the reviewer: why calibrating on both ice volume and area deteriorates the performance. The reviewer is right that the poor results stem from the poor emulation accuracy. From further tests, we concluded that the problem is that we have not originally considered a training set with ice sheet geometries spanning a two-dimensional parameter space designed in such a way to span volume and areas in an optimal way. This would have been a challenging training set to produce because our choice has been to use, for the calibration, ice sheet geometries that have been effectively simulated as output of the ice sheet model. In that sense, we are restricted to what the ice sheet model gives us. Ice volumes and areas are then somehow correlated, but not quite as we explained above, so that the area-volume space is poorly covered. This explains us why, given our design, accounting for both parameters tends to deteriorate the performance of the emulator rather than improving it.

This said, we could improve a bit on the original results. Specifically, the original emulators calibrated on both ice volume and ice area had a reasonably small length scale for the ice sheet parameter. Hence, the ice sheet parameter was in effect not so influential on the emulated temperatures compared to the $CO_2$ forcing and orbital parameter forcing. We performed new experiments with the same correlation length as for the emulator calibrated on a single ice sheet parameter and included these results in the manuscript. Now the transition to a fully glaciated continent occurs for all emulators, but gives only similar results for EMULATOR_12a for any of the calibrations. This is because the relative change in ice area and ice volume between the different prescribed ice sheet geometries is quite similar for the model design in EMULATOR_12a. EMULATOR_12b and EMULATOR_20 also work fine, as expected, when calibrated on a single ice sheet parameter because of the good spread of the ice sheet parameter in the model design.

[revised manuscript text omitted]

3. I am not sure why Section 4 is called 'Bayesian' sensitivity analysis because nothing in the section seems to be particularly 'Bayesian'. There seems to be no consideration on uncertainty in the model parameters in the form of posterior densities, which is typically done in Bayesian analysis. In addition, the authors somehow decided to throw away the emulators and build a new time series model for uncertainty quantification. Is there any particular reason behind this decision? I am also wondering if there is any particular reason that only the first-order autocorrelation is considered here.

**Author's response:** In section 4, the sensitivity of the ice sheet evolution is tested based on Bayesian statistics, i.e. the climate is predicted, taken into account the climate at the previous time step. In that sense, we believe that the use of Bayesian sensitivity analysis is legitimate. We do not investigate the influence of the climate evolution based on different model parameters, but the sensitivity to having prior knowledge about the climate that was simulated one coupling time step before (emulator errors are correlated in the input space). The reason to only include the first order auto-correlation is for simplicity and justified because the first order auto-correlation captures the correlation structure of the emulator well (by design, this is an exponential decay).

*'The additional value of the use of an emulator for coupled ice sheet-climate simulations is that the mean climate predictions come with the estimate of its variance and that two different predictions have a covariance. Here, the uncertainties caused by the emulator variance are explored in order to sample climate trajectories. The covariance between output points given by the emulator is used to update the mean and variance of a climate prediction at a given iteration (e.g. time iteration i) of the ice sheet model, given the climate used at the previous point iteration (i-1). It is then possible to sample the updated distribution. This provides a climate sample at iteration i and the procedure continues to obtain climate samples at iteration i+1 and so forth. The process yields a sample climate trajectory. Strictly speaking, the emulated climate at iteration i is correlated with the outputs at iteration i-1, i-2, etc. and all of them should be used to update the mean and variance at iteration i. However, since the Gaussian process emulator has an exponential decaying correlation function which is short-ranged (in contrast to a power-law), it is expected that the covariance structure of emulated climate trajectories that is associated to the emulator variance is effectively captured by the first-order autocorrelation.'*

The emulator is akin to the emulators shown in section 3, with the emulation of precipitation and January temperature exactly the same way as in the other experiments. Since only the uncertainty of one of the emulated fields can be investigated at once, the parameter most decisive for ice sheet growth is chosen. It appears to be temperature, especially summer temperature and hence uncertainty to January temperature is emulated using a Bayesian approach.

*'So, the emulator is almost identical to EMULATOR_12b, except that covariances are used to sample trajectories around the mean. If the variances are assumed to be zero at all time steps, the mean trajectory already presented in Fig. 9a is approximated (slight difference due to the application of the annual cycle to the January temperature).'*

**Minor Comments**:

1. Lines 204-215: Related to the Major Comment 1 above, I think describing EMULATOR_100a and EMULATOR_100b as 'bad' emulator seems to be weird. I think this part can be improved by clarifying how EMULATOR_100a and EMUATOR_100b are actually designed; otherwise, readers may wonder why the authors decided to use 'bad designs'. Later, in Section 3, they will be surprised by the fact that these 'bad designs' led to better results.

*Author's response:* We tried to clarify what is investigated with the different emulators and what is a 'good' design and a 'bad' design. Clearly, bad designs do not lead to good results and oppositely, good designs (in terms of ice sheet spacing) can only lead to bad results when insufficient ice sheet geometries are included.

Lines 239-240 "Therefore, they might be doing a poor job in reconstructing the simulated temperatures well.": I think 'well' should be deleted.

*Author's response:* The sentence was revised to: *'They have the most extreme insolation values and lay at the edge of the experimental design, which may explain why the emulator does a more poor job at predicting the simulated temperatures.'*

2. Line 259-260: "The notion of ice sheet parameter as an emulator input is introduced in previous studies to be an integer ranging from 1 to the number of ice sheet geometries": The sentence does not make much sense. Please revise.

*Author's response:* The sentence is revised to: *'The ice sheet parameter is a single number representing the shape and area of the prescribed ice sheets. In previous studies (Araya-Melo et al., 2015; Lord et al., 2017), it is defined as an integer, representing the number of different ice sheet geometries.'*

3. I feel that the overall writing quality of Sections 3 and 4 are notably worse than that of the other Sections. Hopefully the authors can improve the texts in the revised version.

*Author's response:* We acknowledge that most comments were related to the model set-up and the model performance. The text has been revised by stating why certain decisions have been made or what we learned from certain decisions. We have tried to improve the clarity of the manuscript by including an additional emulator with 20 prescribed ice sheet geometries, since the number of prescribed ice sheet geometries has a main influence on the emulator performance. We also rewrote the critical paragraphs from section 3, including the results with the additional EMULATOR_20 and gave arguments why the use of the Bayesian sensitivity analysis in section 4 was justified.

**Response to Anonymous Referee #2**

This paper describes a novel method for ice sheet model forcing using a gaussian process emulator, presents new simulations using this method and performs sensitivity analyses. This is a useful contribution that seeks to overcome the limitations of ice sheet climate coupling when performing multi-million year simulations. It is a well thought out study and I recommend it for publication. I have some minor comments to improve the clarity of the article.

**Author's response**: We thank the reviewer for the positive evaluation of the manuscript and the suggestions to improve the clarity.

Minor Issues:
L106: I understand how this approach works with an atmosphere-only GCM, how does this differ for a slab-ocean model? If SSTs are prescribed what is the slab ocean doing? Unless I've misunderstood this, in which case clarification would be useful

**Author's response**: It is clarified in the manuscript why prescribed SST's are needed.

*'…in order to calibrate the corrective heat fluxes from the slab ocean model. These corrective heat fluxes represent the seasonal deep water exchange and horizontal heat transport that is present in the real ocean. The oceanic heat fluxes are exchanged between the atmosphere and the slab ocean model in the mixed-layer, which is 50 m thick in our simulations. This way, realistic sea surface temperatures are simulated for the different climate model simulations.'*

L106: I'm not suggesting doing this here, but for a true simulation of the EOT would two emulators be needed, one with late Eocene and one with early Oligocene SSTs?

**Author's response**: Since the simulated SST's are variable, this is not needed. The SST is only fixed in order to calibrate the corrective heat fluxes.

L118: How is ungrounded ice treated? I imagine there isn't much with the Wilson et al., topography that is being used, but this needs stating.

**Author's response**: Ice shelf formation is included and calculated using the Shallow Shelf Approximation. Ice shelves start to form when the grounding line reaches the coast and the influx of ice from the continent exceeds the ablation (surface ablation and basal melting). A constant basal melt rate of 1 m per year is used in all the simulations. This information is added to the manuscript.

Figure 7: To aid clarity of this figure can you sort the x-axes based on agreement? This would make it easier to compare the different methods.

**Author's response**: Sorting the x-axis would not be beneficial for the understanding of the figure. Each bar on the x-axis represents one experiment from the model design for the 100 experiments going from experiment 1 (xaemaa) to experiment 100 (xaemdv). When the x-axis would be sorted, it would be impossible to see which experiments perform worse than another.

Instead, because another emulator is introduced to increase the clarity of the manuscript and this would lead to 4 (number of emulators) x 3 (ways to calibrate the ice sheet parameter) subplots, we only show the leave-one-out performance for one emulator (EMULATOR_20) with the three different ways to calibrate the ice sheet parameter in the manuscript.

L250: Can the emulator be used to predict a spatially and temporally varying lapse rate?

**Author's response**: This could be done, but in these simulations, the spatially and temporally variable lapse rates are used as they were simulated with the prescribed climate model runs.

L307: Could be worth mentioning that another problem of the single ice sheet parameter (that is common to the matrix method) is that there is no guarantee that ice is growing in the same place in the ice sheet model as is prescribed in the climate model (Figure A2). I.e. a feedback from growing ice on the Antarctic Peninsula could be applied to the Transantarctic Mountains. Is there anyway of overcoming this? E.g. having a regional ice sheet parameter?

**Author's response**: We thank the reviewer for raising this interesting point. It is indeed true that the single ice sheet parameter does not define in which places the ice would grow. However, since the bedrock topography in the coupled ice sheet-climate simulations is the same as in the prescribed climate model runs, ice naturally starts to grow on the highest elevations and the pattern of ice sheet growth is definitely acceptable.

It is hard to overcome this issue and there are reasons why the implementation of a regional ice sheet parameter would not give the right results. Implementing a regional ice sheet parameter would come with introducing another variable that receives a certain value on whether ice is present in that region or not. There is a risk that this regional parameter is again strongly correlated to the ice sheet parameter. Therefore, special care has to be taken in the model design to construct optimal ice sheet parameters with a good spread that are ideally mostly uncorrelated. The best way to overcome the problem of possible ice sheet growth in different regions than the prescribed input ice sheet geometries, is by implementing enough different input ice sheet geometries for the prescribed climate model runs in order that the climate captures the boundaries of many different ice sheets. We have added a discussion on the introduction of regional/multiple ice sheet parameters in the discussion section:

*'A common problem for the emulator and the matrix look-up table method, where the ice sheet parameter is defined by a single number, is that there is no control on the regions where ice starts to grow. The problem can be addressed by describing the ice sheet location and geometry with a vector of several dimensions. In reverse, the definition of this vector and the experiment design have to be thought such as to provide a reasonably orthogonal experiment design,* in order to avoid the issues experienced in this study by attempting to calibrate the emulator both on ice volume and ice area simultaneously. Optionally, ice sheets could be described with additional *variables such as shape factors that are relatively independent on the other ice sheet parameters. We leave the suggestion of creating other ice sheet variables to improve the emulator performance for future work.'*

L310: How much slower is the model with these different coupling timesteps?

**Author's response**: Halving the coupling time step increases the computational time with about 40 %. This information is added to the manuscript.

Figure 14: There are a lot of color-blind unfriendly colors. Here you could just show one of the emulators, rather than two.

**Author's response**: We assume that this remark was based on Figure 16. We changed the figure and only show the annual cycle of the temperature with respect to the January temperatures. The colour palette has also been changed to increase the figure attractiveness.

Figure 17: Can just show one of these subplots as the impact of the timestep already explored.

**Author's response**: Since the behaviour of the ice sheet evolution for the emulator including the estimates of variance is also different when using different coupling time steps, we opted for leaving both figures in the manuscript.

L487: This suggestion of using a direct mass balance calculation comes very late, it might be worth expanding on this point more or removing.

**Author's response**: We decided to remove the suggestion of emulating directly the mass balance for coupled ice sheet-climate simulations.

There are supplementary videos which are not available to review, can these be uploaded to GMD rather than zenodo?

**Author's response**: Yes. This will be done for the revised manuscript.

Typos, etc:
L9: "considering" to "using"

**Author's response**: Done.

L12: "from" to "in"

**Author's response**: Done.

L12: Sentence starting "The sensitivity…" is hard to follow. Suggest rewriting, or adding comma ", and to the coupling time".

**Author's response**: This phrase has been split into two parts as follows:

*'The sensitivity of the evolution of the ice sheet over time is tested with respect to the number of predefined ice sheet geometries the emulator is calibrated on. Additionally, the model performance is evaluated to the formulation of the ice sheet parameter (being either ice sheet volume, either ice sheet area, or both) and to the coupling time.'*

L42: "ran" to "run"

**Author's response**: Done.

L56: "paleoclimate" to "climate during the Pleistocene"

**Author's response**: Done.

L64: need reference for these CO2 changes.

**Author's response**: Two references are added that indicate the large $CO_2$ variations during the late Eocene to early Oligocene: Pagani et al. (2011) and Zhang et al. (2013).

L142: "amount" to "number"

**Author's response**: Done.

L180: (Eq 2.)

**Author's response**: Added.

L234: "Poorly"

**Author's response**: Done.

L235: "ran" to "run"

**Author's response**: Done.

L235: "warm-biased", "cold-biased"

**Author's response**: Done.

L236: remove "locally"

**Author's response**: Done.

L259: "The notion of an"

**Author's response**: This sentence has been rephrased.

*'The ice sheet parameter is a single number representing the shape and area of the prescribed ice sheets. In previous studies (Araya-Melo et a;., 2015; Lord et al., 2017), it is defined as an integer, representing the number of different ice sheet geometries.'*

L286: "with up to" to "by up to"

**Author's response**: Done.

L290: "mostly follows"

**Author's response**: Done

L425: "GP" define or change to "gaussian process"

**Author's response**: GP was defined the first time on line 131.

---

## Author Response (AR2)

**Response to Anonymous Referee**

1.  I still disagree with the authors' argument that their procedure presented in Section 4. They are just computing the conditional means and variances based on the previous time step, which are commonly used by both Bayesians and Frequentists. I think the authors got confused about the distinction between the conditional distribution and the posterior distribution. I strongly suggest that they remove the term 'Bayesian' there. It causes more confusion than clarification.

**Author's response:** We thank the reviewer for his clarification on Bayesian and Frequentists views and changed the title in section 4 from 'Bayesian sensitivity analysis' to 'Uncertainty analysis'.